# Ultrastructural Characteristics of the Juvenile Chum Salmon (*Oncorhynchus keta*) Cerebellum: Interneuron Composition, Neuro–Glial Interactions, Homeostatic Neurogenesis, and Synaptic Plasticity

**DOI:** 10.3390/ijms262211123

**Published:** 2025-11-17

**Authors:** Evgeniya V. Pushchina, Evgeniya E. Vekhova, Eugenia A. Pimenova, Anna V. Akhmadieva, Mariya E. Bykova

**Affiliations:** 1A.V. Zhirmunsky National Scientific Center of Marine Biology, Far Eastern Branch, Russian Academy of Sciences, 690041 Vladivostok, Russia; evekhova@imb.dvo.ru (E.E.V.); epimenova@imb.dvo.ru (E.A.P.); mbykova@imb.dvo.ru (M.E.B.); 2Research and Manufacturing Company “Home of Pharmacy”, 245, Zavodskaya St., Kuzmolovskiy t.s., Vsevolozhsk District, Leningrad Oblast, 188663 Saint Petersburg, Russia; akhmadieva.av@doclinika.ru

**Keywords:** cerebellum, transmission electron microscopy, scanning electron microscopy, chum salmon, interneuron composition, stellate cells, Golgi cells, Purkinje cells, eurydendroid cells, synaptic structure, synaptic contacts of electrotonic types, interneuronal communications, neuro–glial relationships, astrocytes, microglia, glial and non-glial types of precursors

## Abstract

Currently, the problem of climate change on Earth is becoming increasingly urgent. These changes are the reason for the increasingly pronounced adaptive differences in different species of fish. A significant gap in ultrastructural data on the organization of the salmon cerebellum was the main motivation for this study’s microscopic and ultrastructural analyses using transmission and scanning electron microscopy of the cerebellum of juvenile chum salmon *Oncorhynchus keta.* The study of the interneuron composition of the cerebellum showed the presence of stellate cells in the molecular layer, projection Purkinje cells, and eurydendroid cells in the ganglion layer. Large Golgi cells and granular cells were found in the granular layer. The study of the synaptic structure of the molecular layer showed the presence of synaptic contacts of electrotonic and chemical types, which are an important link in interneuronal communications. Most synaptic endings of parallel fibers of the excitatory type in juvenile chum salmon converge onto dendrites of Purkinje cells. Transmission electron microscopy (TEM) study of neuro–glial relationships also revealed a heterogeneous population of astrocytes and microglia in the cerebellum of juvenile chum salmon. Patterns of apoptosis and phagocytosis involving protoplasmic astrocytes were detected. The presence of protoplasmic astrocytes in the cerebellum of juvenile chum salmon contrasts with data reported for zebrafish. The conducted studies allow us to conclude that the homeostatic growth of the cerebellum of juvenile chum salmon can occur according to an uncertain pattern and be mediated by the presence of adult-type neural stem/progenitor cells (aNSPCs). The presence of aNSPCs of glial and non-glial types in the cerebellum of juvenile chum salmon was demonstrated by TEM and scanning electron microscopy (SEM). The discovery of a large population of non-glial aNSPCs in the dorsal matrix zone (DMZ) and granular layer of juvenile chum salmon, as well as stromal cell clusters on the surface of the cerebellar molecular layer, suggests the activity of a neurogenic program in the brain of juvenile chum salmon that is mainly active during embryonic stages in other vertebrate species. The phenomenon of embryonization in the cerebellum of juvenile chum salmon is determined by the presence of non-glial aNSPCs, which contribute to homeostatic growth.

## 1. Introduction

The cerebellum is a suprasegmental integrative center of the brain and is responsible for the coordination of movements and the implementation of fine motor programs. Despite the existence of fundamental studies of the structural organization and connections of the mammalian cerebellum, the essential aspects of its involvement in behavioral activity and cognitive functions are still being studied. The cerebellum of salmon consists of three parts: the *corpus cerebelli* (CC), the *valvula cerebelli* (VC), extending under the lobes of the *optic tectum*, and the vestibulo-lateral (caudal) lobe [1,2]. The cerebellar cortex occupies the largest part and forms an unpaired dorsal protrusion of the rhombencephalon. The VC is a novel structural formation in the evolutionary context of the cerebellum of ray-finned fishes and represents a rostral continuation of the cerebellar cortex into the tectal ventricle [3]. The vestibulo-lateral lobes of the cerebellum have connections with octavolateral neurons and include the caudal lobe and the paired *eminencia granularis* (GrEm) [4,5,6,7].

Pacific chum salmon is an interesting object for the study of cerebellar plasticity, as it contains a large number of precursors of various types [8]. The results of IHC studies showed that the cerebellum of juvenile chum salmon is characterized by high proliferative activity and contains proliferating cell nuclear antigen+ PCNA+ cells of various morphologies and localization. IHC study of aNSPCs in the cerebellum of juvenile chum salmon showed the presence of both glial fibrillary acidic protein and vimentin (GFAP+, Vim+) and non-glial nestin (Nes+) precursors of the adult type [8]. In previous studies, it was found that in the cerebellum of juvenile chum salmon, the glial and non-glial precursors of adult types are morphologically heterogeneous. IHC analysis revealed cells of various sizes and morphologies, as well as migrating cell populations. For further verification of cell types, more in-depth studies are needed to characterize the identified cell types in more detail. For this purpose, the TEM and SEM methods were applied in the present study. The need for an expanded ultrastructural study is associated with further detailed verification of cells in the granular layer (GrL), as well as an extended characterization of cells in the molecular layer (ML). An ultrastructural study of neuro–glial relationships, in light of protoplasmic astrocytes found in juvenile chum salmon, is also necessary to understand the role of these cells in the metabolism of Purkinje neurons and eurydendroid cells (EDCs).

The neuronal organization of the cerebellum in teleosts has been previously studied by silver impregnation methods [3,9,10,11] and TEM methods [10,11,12]. Ultrastructural analysis of neuronal organization and differentiation allows us to formulate recently identified features and developmental trends of the neuronal structures of the fish cerebellum. These data are of particular interest in the context of adult neurogenesis, since the cerebellum of juvenile masu salmon *Oncorhynchus masou* [13] and trout *O. mykiss* [14], like other parts of the salmon brain, is capable of forming new cells during growth. New cells also form after traumatic injury to the cerebellum in masu salmon *O. masou* [15,16] and chum salmon *Oncorhynchus keta* [17].

Cerebellar cells originate from various sources, in particular from the neural stem cells (NSCs) located in the dorsal parts of the molecular layer (ML), parenchyma, GrEm, and granular layer (GrL) [13,14,16]. However, to date, there is no consensus on the origin and morphogenesis of most cerebellar cells and the degree of their homology to mammalian cells; for example, basket cells have not been found in teleosts.

It has been reported that the cerebellum of salmon fish has an outer ML containing cells of stellate morphology (SC), dendrites of Purkinje cells (PCs), and parallel fibers (pfs) originating from granular cells (GrCs) [2]. The cells of the ganglion layer (GL) are represented by large projection Purkinje cells, which form a single layer in salmon fish. In the GrL, the main cell type is small granular cells (GrCs), among which there are large Golgi cells (GCs) [10,11,18]. In the opisthocentrid fish *Pholidapus dibowskii*, large cells in the GrL of the cerebellum, with bipolar morphology resembling Lugaro cells of higher vertebrates, have been described [19]. PCs in ray-finned fish form connections only within the cerebellum [2], whereas external extracerebellar projections are formed by EDCs located between PCs [20,21,22]. Because of the presence of the EDCs instead of the deep cerebellar nuclei, Purkinje cells do not project beyond the cerebellar cortex but function as interneurons with short axons terminating within the GL [23,24].

The details of the neurochemical and ultrastructural architecture of the various interneuron types in the fish cerebellum have not been adequately addressed to date. Information on the features of the synaptic structure and the pattern of the intercellular contacts formed in the cerebellum of salmonids is also scarce, despite the fact that the connections between the cerebellum with various areas of the brain in juvenile trout *O. mykiss* [2] and masu salmon *O. masou* [25] have been studied using diffusion transport methods. Studies conducted on different species of salmonids have proven to be important for understanding the cellular organization and morphogenesis of the cerebellum in teleosts [26]. In particular, the connections of the cerebellum in masu salmon and trout have been studied experimentally, including in the context of the organization of cholinergic systems [25,27]. A comprehensive analysis of the cerebellar connections in salmon using a lipophilic dye (DiI), capable of diffusing along cell membranes in a fixed brain, made it possible to track the similarity in the organization of the main extracerebellar projections to various parts of the brain [2,25].

Morphological portraits of neurons and glial cells of the cerebellum illustrate in detail both general and specific features for each type. These features are determined by clear structural parameters, including the size of the cell body; layer-by-layer and spatial localization of perikarya; size, density, and distribution pattern of the dendritic spine apparatus; comparative geometry of dendritic fields; polymorphism of axon arborizations; and the type of synaptic terminals [28]. The aim of this work was to study the ultrastructural organization of the cerebellum of juvenile chum salmon *O. keta* in the context of interneuron composition, neuro–glial relationships, homeostatic neurogenesis, and synaptic plasticity.

## 2. Results

### 2.1. Morphological Organization of the Cerebellum of Juvenile Chum Salmon Oncorhynchus keta

#### 2.1.1. Organization of the Corpus Cerebellum

The features of the structural organization of the *corpus cerebellum* (CC) are shown in Figure 1. The CC of juvenile chum salmon has a typical organization, including the ML, GrL, and GL (Figure 1A). Figure 1 shows semi-thin sections demonstrating the morphology of the CC cells. The dorsal part of the CC contains the DMZ, which is the main neurogenic region in juvenile chum salmon (Figure 1B). This area is heterogeneous in its cellular composition and includes a few elongated, lightly stained cells of the neuroepithelial type arranged along the periphery of the DMZ and numerous darkly stained, star-shaped cells, which are aNSPCs and occupy the central part of the DMZ (Figure 1B). There are undifferentiated, densely stained cells of small size in the dorsal and ventrolateral parts of the DMZ (Figure 1B). The morphological parameters of the DMZ cells are presented in Table 1.

The features of the organization of the cerebellar layers in juvenile chum salmon vary in different areas; however, the general structure remains unchanged (Figure 1C). In the basolateral zone of the juvenile chum salmon cerebellum, the GL cells often form clusters within which PCs with oval perikarya (Figure 1D) and elongated cells with bipolar morphology (Figure 1D) can be morphologically distinguished. The latter apparently correspond to EDCs. In addition, morphological analysis showed the presence of macroglia cells, astrocytes with short processes ending in end-feet on the bodies of PCs (Figure 1C,D).

Another type of macroglia, radially directed Bergmann glia, was also found in the cell complexes of the ganglion layer of the cerebellum (Figure 1D). In the basolateral region of the CC of juvenile chum salmon, complexes of large PCs surrounded by clusters of granular cells were also identified, among which the darkest stained single cells were aNSPCs (Figure 1C, red rectangle). In the ML of the basolateral region, stellate cells (Figure 1C, black dotted rectangle) were relatively rare (Figure 1C, crimson arrow).

Another type of large multipolar Golgi cells was identified at the border of the ML and GrL (Figure 1C,D, pink arrow). Morphological examination of the GrL showed the features of patterning of the cells within the GrL (Figure 1C,D, red dotted ovals). It was found that the cells of the GrL form clusters, within which morphologically heterogeneous cells, separated by fibers, are grouped (Figure 1D). The densely stained, irregularly shaped type I cells represented the aNSPCs, while the larger, weakly stained, rounded cells represented the glutamatergic granule cells (Figure 1E).

The basal part of the molecular layer contained patterns of sprouting unmyelinated fibers (Figure 1F, red rectangle), fragments of microvessels (Figure 1F, white arrows), Bergmann glial fibers (Figure 1F, blue arrows), and single aNSPCs (Figure 1F, green arrows).

#### 2.1.2. Organization of the *Valvula Cerebelli*

Morphological analysis of the *valvula cerebelli* (VC) of juvenile chum salmon, located rostrally to the CC, showed the presence of all the main layers identified in the CC (Figure 2A–C). In the rostro-lateral part of the VC, the GL, represented by large PCs (orange arrows) and EDCs (light green arrows), was identified inside (Figure 2A), and was bounded outside by cells of the GrL (Figure 2A). Large neurons with bipolar morphology (black arrow) were also identified in this area, located at the border between the GrL and GL (Figure 2A).

In the GrL of the VC, tendencies towards cluster organization of heterogeneous cell types were observed (Figure 2A,B, red dotted ovals), similar to that in the CC. In the portion of the GrL facing the inside of the intertectal ventricle, the clusters were larger compared to those in the dorsal layer (Figure 2A,B). In the medial part of the VC, patterns of PCs distribution were revealed, with some PCs being in contact with large protoplasmic astrocytes (Figure 2C). In the central part of the VC, cells of the ML were present, containing typical forms of astrocytes and aNSPCs (Figure 2C). SCs were not identified. In the ML, individual small cells were found in the supraganglionic region or in the parenchymatous zone (Figure 2C, pink arrows).

#### 2.1.3. Organization of the Granular Eminences

Granular eminences (GrEm) were located in the ventrolateral zone from the CC of juvenile chum salmon and included a centrally located GrL and a peripherally positioned ML (Figure 3A). The organization of GrEm was largely similar to that of the CC (Figure 3B). However, clusters of GrCs were smaller and were represented mainly in the CC (Figure 3B, red dotted ovals), fibers (Figure 3A, green arrows), macroglial cells (Figure 3A, scarlet arrows) and Golgi cells (Figure 3A,B, pink arrows), which were quite common. In general, the distribution density of granular cells in the GrEm was lower than in the CC. Along with small clusters, a diffuse type of distribution of GrCs predominated.

In the ventrolateral zone of the GrEm, at the border of the ML and Gl, fibers of different directions were observed, representing the lateral cerebellar peduncles (Figure 3C, green arrows). GCs of various sizes were frequently encountered (Figure 3C, pink arrows); their morphometric parameters are presented in Table 1. Small clusters with densely stained dendritic aNSPCs were found at the border of the molecular and granular layers (Figure 3D, in red dotted rectangles). Another population was represented by rounded, transparent, weakly stained cells corresponding to the non-glial type of aNSPCs (Figure 3D, red arrows).

The density of aNSPCs clusters in the ventrolateral part of the GrEm increased markedly (Figure 3E, yellow arrows). These cells had glial-like morphology and represented aNSPCs. The morphological parameters of the GrEm cells are presented in Table 1. In the ventromedial zone of the GrEm, clusters of unmyelinated fibers of varying diameters were detected (Table 1, Figure 3F, red arrows). These fibers formed sprouting bundles that constituted the cerebellar peduncles.

### 2.2. Ultrastructural Organization of the Cerebellum of Juvenile Chum Salmon

#### 2.2.1. Molecular Layer Cells

The ML of the juvenile chum salmon cerebellum included stellate cells (SCs) located in the upper third and deeper layers (Figure 4A). The sizes of the SC somata are presented in Table 1. The basal part of the SCs contained a large nucleus with clumped heterochromatin (Figure 4A,B, Table 1). The cytoplasm of the SCs was of medium density, with elongated large and small mitochondria evenly distributed in it (Figure 4A–C). Numerous secretory granules were found in the basal part of the SCs (Figure 4A). The SCs had 4–6 short processes that were not fully visualized in the sections; only their points of origin could be determined (Figure 4A).

Large asymmetric synapses containing synaptic vesicles with glutamate were identified at the contact areas with PC dendrites (Figure 4B). Synaptic terminals identified in the ML had variable morphology, with their sizes ranging from 500 to 950 nm (Figure 4B,C). Such nodular-type terminals were quite densely distributed in the ML, and some were comparable in size to the SCs (Figure 4C). Imaging studies indicate that parallel fibers (pfs) provide input to the SCs, which extend axons parallel to the pf and, in mice, form inhibitory synapses with PC dendrites to modulate pf input [29].

In the lower third of the ML, fragments of large cells were identified, apparently being proximal fragments of the dendritic trees of PCs (Figure 4B–D). Several synaptic endings, including those of the electrotonic type, converged on the PC bodies (Figure 4D). Multiple excitatory synaptic contacts, the presence of mitochondria and vacuoles, and cytoskeletal components (Figure 4D) distinguished such PC formations from other cellular and fibrous structures in the ML. The PCs were characterized by the formation of extended ascending dendrites (Figure 4D, green arrows); some cells were located very close, sometimes in contact (Figure 4D). Along with interneurons, oligodendroglial cells involved in the formation of myelin fibers were identified in the ML (Figure 4E). Such cells had few processes, but were distinguished by polar invagination of the cytoplasm, which characterized the early stages of myelin fiber formation (Figure 4E). The synaptic structure formed on the surface of the PCs was quite diverse; in some cases, it was possible to identify the components of chemical synapses: a presynaptic terminal, a synaptic cleft, and a postsynaptic component (Figure 4F).

#### 2.2.2. Morphological Heterogeneity of Synaptic Structures

The morphological heterogeneity of synaptic structures revealed in the caudal region of the ML of the juvenile chum salmon cerebellum allowed identification of different types of synaptic terminals converging on the dendrites of PCs and SCs (Figure 4C and Figure 5A). It is known that one of the most prominent synapses of the cerebellum is formed in the ML of the zebrafish cerebellum: the synapse between pfs and PCs, which provides excitatory input to the PCs. Studies on zebrafish have shown that pfs provide input signaling to SCs, whose axons extend in the same plane as pfs [30].

A characteristic feature of the organization of the dendritic processes of the PCs, contacting the synaptic terminals, is the presence of numerous mitochondria, which indicates a high level of their energy metabolism (Figure 5A). The smallest axosomatic contacts of the electrotonic type, 430–630 nm in size, had an oval or elongated morphology (Figure 5A). There were up to five such terminals on the bodies of the SCs. They were typically asymmetric and did not contain a postsynaptic component. Another type of contacts was represented by larger terminals up to 1.2 μm in size, which were characterized by the presence of a presynaptic structure, a synaptic cleft, and a postsynaptic component (Figure 4B,F and Figure 5A,B). In some cases, paired large synaptic terminals with a pronounced pre- and postsynaptic structure were revealed on the bodies of the PCs in the ML (Figure 5B).

Synaptic structures with complex, convoluted T-shaped morphology, containing individual mitochondria, were identified within the dendritic bouquets of PCs and EDCs (Figure 5C). Such structures had large synaptic terminals of complex shapes, containing mitochondria that were often connected via fibers to smaller club-shaped endings (Figure 5C). The presence of vesicles with neurotransmitter could be traced in all types of synaptic structures contained in climbing fibers (cfs) (Figure 5C, yellow arrows). Single, elongated terminals, 1.3 μm in size, with pre- and postsynaptic structure were identified (Figure 5D). The presence of numerous synaptic terminals, with a high density distribution in the ML, converging on cfs (Figure 5E) and forming contacts in the extracellular space of ML (Figure 5F), indicates a significant complexity of the synaptic structure regulating the cerebellar impulse in the molecular layer and a complex pattern of synaptic connectivity in the juvenile chum salmon cerebellum.

#### 2.2.3. Ultrastructural Organization of Climbing Fibers and Neural Stem Progenitor Cells

Climbing fibers are afferents emerging from the internal olives and directed to the ML [31]. In zebrafish, cfs contact the PCs in the area of the basal dendrites, whereas in mammals, cfs reach the distal branches of the PCs’ dendritic tree [32]. Cfs in the fish brain primarily transmit sensory and visual information, which is necessary for coordination during hunting, exploration, or escape maneuvers and, thus, are of behavioral importance. Sensory signaling transmitted by cfs informs the cerebellum of temporary changes in movements [32]. Based on this stimulus, oculomotor processing occurs in specific regions of the GL, which likely represent behavioral modules responsible for sensorimotor integration and motor learning in the cerebellum [32,33]. To date, no data have been found on the stereoscopic structure of the cfs in fish; however, such data could provide interesting and useful information on the spatial organization and integration of this type of fiber.

A scanning electron microscopy (SEM) study of the ML of juvenile chum salmon revealed a complex and branched structure of liana-like afferents with varicose dilatations and terminal thickenings (Figure 6A). An individual cfs has an uneven thickness, and thickenings of different volumes are observed at the branching sites (Figure 6B). In general, the surface over which cfs spread in the cerebellum also has a complex and heterogeneous relief, including numerous synaptic structures (Figure 6C). When magnified, these synaptic areas have an appearance of a multidimensional network with numerous spherical thickenings (Figure 6D).

Spherical cells, identified in the superficial dorsolateral regions of the ML, were other objects for stereoscopic ultrastructural study. Previously, we showed that in the trout *O. mykiss* [14], masu salmon *O. masou* [13,15], and chum salmon *O. keta* [16], such cells are labeled with PCNA and also contain vimentin and nestin [14,16], i.e., they are aNSPCs. SEM was used to detail their spatial ultrastructure. The results of the study showed that these cells are often located near erythrocytes and are fixed on the surface of the ML by microfibrils (Figure 7A). The surface microcytosculpture of such cells was heterogeneous and had a bumpy relief (Figure 7B). Surrounded by the extracellular matrix of the pial membrane, aNSPCs were often adjacent to biconcave erythrocytes with a smooth surface (Figure 7C). As a result of proliferation, aNSPCs often formed paired cell clusters (Figure 7D). In some cases, small clusters with several aNSPCs were identified (Figure 7E). Diffuse patterns of NSC distribution often demonstrated their association with the surface matrix, as well as erythrocytes (Figure 7F). Overall, the SEM data suggest that spherical cells with a superficial localization in the body of the cerebellum can be classified as aNSPCs.

#### 2.2.4. Ultrastructural Organization of the Dorsal Matrix Zone

The dorsal matrix zone (DMZ) of the juvenile chum salmon cerebellum is the main neurogenic region in adult animals (Figure 1B). Semi-thin sections revealed the heterogeneity of the DMZ’s cellular composition, showing elongated neuroepithelial cells (NECs) at the periphery (Figure 1B). Data of ultrastructural analysis confirmed the presence of single NECs located in the peripheral parts of the DMZ (Figure 8A,B). Another type was rounded, light-stained cells (Figure 8A, inset), with body sizes of 7.71 ± 0.45 by 6.06 ± 0.53 μm and a rounded, centrally located nucleus, measuring 6.65 ± 0.36 by 5.46 ± 0.54 μm (Table 1). We consider cells of this type as adult-type non-glial aNSPCs, which have a clear cytoplasm and a clear nucleus filled with reticular euchromatin. Such cells were found in several areas of the cerebellum, in particular, in the dorsal part of the cerebellum in the DMZ (Figure 8A,C), in the ventrolateral areas of the cerebellum (Figure 1E), and in the superficial layers of the dorsal and dorsolateral areas (Figure 7A–D).

The third type was numerous darkly stained, stellate-shaped glial aNSPCs, occupying the central part of the DMZ (Figure 8A,B). This population of aNSPCs was heterogeneous and was represented by small cells and larger types of progenitors (Table 1). According to the classification of Lindsey and colleagues [34], these aNSPCs corresponded to type III, had electron-dense cytoplasm, a large, irregularly shaped nucleus, and short processes (Figure 8B). Type III cells formed clusters in the peripheral areas of the DMZ (Figure 8C), where they were adjacent to young neurons and/or neuroblasts. Elongated, large cells with bipolar morphology, corresponding to pericytes or astrocytes (Figure 8D), surrounded by numerous myelinated fibers, were identified in the DMZ.

Analysis of the distribution of glial and non-glial aNSPCs in the ventrolateral part of the cerebellum showed that non-glial precursors of type III formed large clusters in the parenchyma of the ML (Figure 8E), as well as at the border of the GrL (Figure 8E). In the granular layer, non-glial aNSPCs might alternate with clusters of type III glial aNSPCs (Figure 8F), where patterns of intercellular localization of secretory granules and myelinated fibers were also revealed (Figure 8F). Overall, the TEM results, in combination with SEM data, allow us to conclude that two types of precursors were present in the superficial areas of the juvenile chum salmon cerebellum, as well as in the GrEm: rounded or spherical ones, forming clusters in the ML parenchyma, and on the surface of the dorsal zone of the cerebellum, belonging to the non-glial aNSPCs. Glial aNSPCs, with an irregularly shaped nucleus and uneven edges, located in the parenchyma of the DMZ and the GrL of the CC, represented another type of precursor. Thus, two types of aNSPCs involved in homeostatic growth of the cerebellum were demonstrated using TEM and SEM in the juvenile chum salmon cerebellum.

### 2.3. Ganglion Layer

Purkinje cells (PC) in the juvenile chum salmon cerebellum form a heterogeneous population, with varying morphological and ultrastructural parameters (Table 1). PCs in the dorsolateral region of the cerebellum had oval or round bodies, with sizes ranging from 18 to 32 μm (Figure 9A, Table 1). PCs formed the ganglion layer (GL), with a distance of 5–10 μm between them (Figure 9A). From the side of the ML and GrL, the GL cells were bordered by fibers of the infraganglionic plexus, which was dominated by multidirectional myelinated fibers (Figure 9A). The neuropil of the infraganglionic plexus was formed by ascending afferents of granular cells (GrCs), forming a system of pfs, converging on the PC dendrites in the upper third of the ML, as well as axon collaterals of the PCs and EDCs. Astrocytes of the protoplasmic type were identified near the PC bodies (Figure 9A), and microglial cells were found in the intercellular space of the GL (Figure 9A). Condensed nuclei and dense apoptotic bodies in the form of crescents (Figure 9A) were localized in the area of the infraganglionic plexus, indicating constitutive apoptosis processes in the cerebellum.

The nucleus of type II PCs was located in the center, large, and contained 1–3 nucleoli shifted to one of the poles of the cell (Figure 9B). The nucleus was predominantly composed of euchromatin, with small fragments of heterochromatin occasionally observed (Figure 9B). The cytoplasm was abundantly vacuolated and contained numerous mitochondria of various shapes and sizes (Table 1). In type I PCs with oval body morphology, the nucleus typically contained reticular light euchromatin with weakly expressed nucleoli (Figure 9B). Type II PCs with a round body, 22 μm in size, had a nucleus with irregularly shaped euchromatin, two larger nucleoli, and small fragments of heterochromatin scattered within the nucleus (Figure 9C, Table 1). The cytoplasm vacuolization and mitochondrial distribution density in rounded PCs were similar to those in oval-shaped cells. The rounded PCs of the lateral area were ectopic in the GrL and were often surrounded by type IV GrCs, whereas the fibers of the infraganglionic plexus were localized dorsally (Figure 9C). In this area, vessels with intra-localized agranular macrophages were found, measuring 10 by 4 μm in size with an amoeboid morphology (Figure 9C). The cytoplasm of the macrophages was dark, homogeneous, and devoid of organelles, while the nucleus was irregularly shaped with heterochromatin (Figure 9C).

Glial type IV aNSPCs in GrL had a body measuring 5.67 ± 1.13 by 3.81 ± 0.42 μm, irregular or oval in shape, with dark cytoplasm, practically devoid of organelles (Figure 9C,D Table 1). The nucleus was light, irregular in shape, contained coarsely lumpy heterochromatin (Figure 9C,D). Some of the type IV aNSPCs were in a state of proliferation (Figure 9C,D). Along with the patterns of GrCs proliferation, patterns of axonogenesis and constitutive fiber formation were revealed in the region of the infraganglionic plexus (Figure 9C,D), indicating the processes of homeostatic neuro- and neuritogenesis in the cerebellum. Bundles of myelinated fibers in the region of the GL often contained thick myelin sheaths and mitochondria (Figure 9D).

In some cases, initial fragments of unbranched dendrites (Figure 9E), surrounded by multidirectional myelinated fibers of the extraganglionic plexus, were identified in the PCs. On the bodies of the PCs, contacts with fibers of the electrotonic type, as well as the ends of cfs, were revealed (Figure 9E).

In the GL, we identified a granulocyte measuring 7.2 by 3.3 μm in size, containing large vacuoles in dense cytoplasm, granules, and a dense homogeneous nucleus (Figure 9E). The granulocyte was surrounded by a thin vascular membrane, which indicated its hematogenous nature. Type III PCs were surrounded by multidirectional fibers of the high-density infraganglionic plexus. An individual PC was surrounded by 40–50 fibers (Figure 9E). Ventrally, the PCs bordered the GrL, where GrCs corresponding to type IV glial aNSPCs were determined, as well as dense cells with processes and a dark, homogeneous nucleus of irregular shape, classified as type III according to the Lindsey’s classification [34]. Along with type IV cells, type III cells were related to glial aNSPCs. Fragments of apoptotic cells were found in the infraganglionic plexus region (Figure 9E).

The fourth population of PCs was found in the ventrolateral region of the CC and was represented by cells having a pear-shaped soma with a thick dendrite extending apically into the ML (Figure 9F). The cytoplasm exhibited a distribution of vacuoles and mitochondria similar to that of other PC types.

The nuclei of these PCs were bean-shaped or oval, containing euchromatin, with small heterochromatic fragments and a single nucleolus, located in the center (Figure 9F). At the basal pole of the type 4 PCs, in the region of the axon hillock, filiform fibrils of actin filaments were identified (Figure 9F, red dotted box). At the apical part of the cell soma, cf endings were found containing mitochondria with a tortuous, ribbon-like morphology (Figure 9F). Near the PCs, neurons with a greater diameter of 10.8 μm and a lesser diameter of 6.1 μm were identified, having a large nucleus and multipolar morphology, which might correspond to EDCs (Figure 9F, Table 1).

### 2.4. Granular Layer

The GrL is the deepest layer of the cerebellum, located ventrally to the GL. Several cell types were found in the GrL of the cerebellum: small granular cells (GrCs) and large Golgi cells (GCs) (Figure 10A–F and Figure 11A–C). Ultrastructural analysis of the GrCs of the cerebellum showed differences in the morphological structure of GrCs, which formed a heterogeneous group of aNSPCs, as well as glutamatergic neurons, which formed pfs branching in a T-shape in the ML.

#### 2.4.1. Granular Cells

In the dorsomedial part of the cerebellum, the neurons of the GrL were represented by small cells with somata sizes of 6.19 ± 0.71 μm and 3.96 ± 0.53 μm (greater and lesser diameters, respectively) (Figure 10A, Table 1). GrCs had large, rounded or oval light nuclei containing constitutive lumpy heterochromatin, distributed along the periphery of the nucleus (Figure 10A). The cytoplasm of GrCs was usually scanty, light, and foamy; it contained vacuoles and mitochondria of various sizes and enveloped the nucleus, forming areas of somato-somatic adjacency in some places (Figure 10A).

Among the GrCs, there were more elongated cells measuring 5.67 ± 1.13 µm and smaller ones measuring 3.81 ± 0.42 µm, with an elongated nucleus of irregular shape, finely lumpy heterochromatin, and a narrow rim of cytoplasm devoid of organelles (Figure 10A). These cells were type IV glial aNSPCs according to the Lindsey’s classification [34]. The ratio of the numbers of GrCs to type IV aNSPCs was 1:2. In deeper layers, cells with electron-dense, homogeneous nuclei of irregular shape and a narrow rim of dark cytoplasm devoid of organelles were identified (Figure 10B). We categorized such cells, according to the Lindsey’s classification [34], as type III glial aNSPCs, which were another type of precursor found in the granular layer of the cerebellum.

Along with GrCs and precursors of types III and IV, myelinated and unmyelinated fibers were found in the GrL (Figure 10A,B). In the lateral part of the CC, in the GrL, mixed diffuse patterns of GrCs and type III and IV precursors were observed, and these areas typically lacked aNSPC clusters (Figure 10C). In the basolateral region of the GrL, the ratio of GrCs and precursors of type IV changed to 1:1 (Figure 10D). A tendency toward clustering of type IV aNSPCs became pronounced, small clusters appeared, and the diffuse distribution of these cells changed to a cluster type (Figure 10D).

In the mediobasal zone, similar clusters, defined here as constitutive adult-type neurogenic niches, were formed by type III aNSPCs (Figure 10E). Interestingly, branching and terminals of the cfs could be seen near the neurogenic cluster (Figure 10E). Figure 10F shows a mixed adult-type neurogenic niche formed by type III and IV aNSPCs at higher magnification.

#### 2.4.2. Golgi Cells

Golgi cells (GCs) are large GABA-ergic interneurons of the GrL. In the juvenile chum salmon cerebellum, individual GCs were identified in different areas of the GrL. The GCs were surrounded by type IV aNSPCs and were often found in the central part of the CC of juvenile chum salmon; the sizes of their somata were 18.3 ± 0.81 μm and 13.02 ± 0.95 μm, and the nuclei were 12.54 ± 0.98 μm and 7.6 ± 0.95 μm (greater and lesser diameters, respectively) (Figure 11A, Table 1). The nucleus was large, slightly elongated, and located in the center with reticular euchromatin. Lumpy accumulations of heterochromatin were distributed throughout the nucleus (Figure 11A,B). At the basal pole, the nuclear envelope was clearly visible (Figure 11B), and at the apical pole of GCs, numerous thin actin fibrils were identified (Figure 11B, red inset). The cytoplasm contained a large number of vacuoles and mitochondria of various shapes and sizes (Figure 11A,C). The rough endoplasmic reticulum (RER) was highly developed in the cytoplasm, with numerous ribosomes concentrated on RER cisterns (Figure 11B,C), and individual ribosome clusters localized in the cytoplasm (Figure 11C, blue arrows). In the apical part of GCs, the Golgi apparatus (GA) with a system of intracellular cisterns and vesicles was identified (Figure 11C, pink dotted inset).

Large protoplasmic astrocytes with soma dimensions of 13.9 by 9.6 μm and nuclei measuring 11.3 by 8.4 μm (greater and lesser diameters, respectively) were also found in the GrL (Figure 11D). These cells had dense cytoplasm with few vacuoles located at the periphery and a large, centrally located nucleus (Figure 11D). Unlike GCs, the astrocyte nucleus was poorly delineated and contained a nucleolus and clumped heterochromatin (Figure 11D). The dense cytoplasm of the astrocyte contained numerous ribosomes on RER cisterns (Figure 11D); mitochondria, on the contrary, were sparse. Thick processes were identified in the astrocyte, and some of them could be traced over a considerable distance (Figure 11E). Astrocyte processes formed contacts with type IV aNSPCs (Figure 11E). Apoptotic patterns of granule cell, in the form of end bodies, were detected near the astrocyte (Figure 11D,E). The intercellular space of the GSs contained numerous granules and sprouting fibers, and was significantly depleted of cells compared to other areas of the GSs (Figure 11D). The astrocyte cytoplasm was filled with ribosomal complexes and was distinguished by a high electron density. Actin fibers, polyribosomal complexes, and electron-lucent mitochondria could be seen in the astrocyte processes (Figure 11E). Signs of destruction were visible in the area of the supposed contact between the astrocyte and aNSPCs (Figure 11E).

### 2.5. Cerebellar Glial Cells

#### 2.5.1. Macroglia

An interesting finding was that some PCs ectopic in GrL formed contacts with protoplasmic astrocytes (Figure 12A). Astrocytes with a cell body size of 4.9 by 2.4 μm had a dark, electron-dense cytoplasm containing mitochondria and single vacuoles (Figure 12B). The nucleus in such cells was not outlined. In the area of contact with PCs, the space between the astrocyte and the neuron was almost absent. However, on the side of the astrocyte plasma membrane, clusters of (structural or receptor) proteins were found, located in the area adjacent to the PCs (Figure 12C). On the side of the PCs, a compaction adjacent to the plasma membrane was also found (Figure 12C, red arrowhead). Thus, large PCs of juvenile chum salmon formed contacts with protoplasmic astrocytes, which is generally not typical for fish. Electrotonic contacts were found in PCs and thick myelinated cfs (Figure 12E). On the side of the PCs, polyribosome clusters, numerous vacuoles, and protein complexes were observed in the contact zone (Figure 12F).

In the GL and GrL of the cerebellum, we found macroglial cells, which were a type of protoplasmic astrocytes (Figure 9 A and Figure 12A). Such cells were usually adjacent to the PCs, the projection neurons of the GL, which were also sometimes located in the GrL. Nevertheless, the population of astrocytic glia in the juvenile chum salmon cerebellum was characterized by morphological heterogeneity. In particular, protoplasmic astrocytes differed in body size, morphology of nuclei and their number, and morphology of processes. Since the issues of plasticity are considered in the present work, we do not rule out that protoplasmic astrocytes represented different subpopulations, which are, possibly, functional analogs of the proinflammatory and anti-inflammatory phenotypes of astrocytic glia in the mammalian brain.

In the GL, we also identified several astrocytes surrounding the PCs (Figure 9A). Such astrocytes were small in size (Table 1), with short, thick processes, dense cytoplasm, and an irregularly shaped nucleus (Figure 9A). Some astrocytes had two irregularly shaped nuclei per cell (Figure 9A).

Large astrocytes (10.69 ± 0.43 by 4.33 ± 0.26 μm) were found at the border of the GL and GrL (Figure 13A). These astrocytes had a large, jagged nucleus (Figure 13A,B) and dense cytoplasm with a developed protein-synthesizing complex of ribosomes and numerous mitochondria. Multiple processes extended from the astrocyte body, some reaching considerable length (2.5 μm) and diameter (150 nm) (Figure 13A). At one of the astrocyte poles, the processes surrounded membrane-bound vesicles, measuring 250–300 nm in diameter (Figure 13A), and formed endocytic complexes (Figure 13A, green inset). Since endocytic patterns were detected in the area of astrocyte–neuron junction, we do not rule out the possibility that this type of astrocyte is specialized in the elimination and utilization of damaged neurons. Astrocytes of this type formed contacts with cfs (Figure 13A, magenta arrowheads). Zones of secretory granules were detected near the astrocytes (Figure 13A, red dotted rectangle). Their granular content could be extracellular glutamate, which the astrocytes absorb for neutralizing and converting into neutral glutamine. Similar patterns of extracellular glutamate utilization were also found in other areas of the GL and infraganglionic plexus (Figure 13C). Some astrocytes, in addition to processes with pseudopodial morphology, showed typical end-feet converging on myelinated fibers (Figure 13C, orange arrowhead). Overall, this type of astrocyte can be characterized as anti-inflammatory with endocytic activity.

Another astrocyte type, with less branched processes, measuring 10 by 5 μm in diameter (greater and lesser diameters, respectively), was also localized among myelinated fibers, forming tangential contacts (Figure 13D, orange arrowheads) and presumably corresponded to mammalian fibrous-type astrocytes. These astrocytes were characterized by two nuclei, were branched and irregular in shape, and occupied most of the cell volume (Figure 13D). The dense cytoplasm of fibrous astrocytes was filled with polyribosomes (Figure 13D, white arrowheads).

In the GL, astrocytes were found adjacent to the EDCs (Figure 13E, yellow arrow). In contrast to other described forms of astrocytes found in the juvenile chum salmon cerebellum, these astrocytes clearly had two nuclei (Figure 13F, arrowheads). This astrocyte type morphologically resembled astrocytes with incomplete division. We suggest that this pattern arose from asymmetric mitoses in which cytoplasmic division was not completed. Typical end-feet were identified in these protoplasmic astrocytes (Figure 13F, orange arrowheads). Interestingly, local, as well as larger, areas containing secretory granules of unknown origin were identified in the GL (Figure 13E, red dotted ovals). We speculate that these granules may contain metabolic extracellular glutamate excreted by pfs.

#### 2.5.2. Microglia

In addition to the heterogeneous population of astrocytic macroglia, resident microglial cells were identified in the juvenile chum salmon cerebellum (Figure 14A). Resident microgliocytes were cells of the immunological series with a branched morphology of processes, the greater diameter of the body of which corresponded to 2.7 μm, and the lesser was 1.78 μm (Figure 14B). We identified branched (ramified) microgliocytes localized near GrCs, having short, highly branched processes, dark granular cytoplasm and a centrally located nucleus (with a greater diameter of 1.2 μm and a lesser diameter of 1.1 μm) with large lumpy heterochromatin (Figure 14B). The branched processes of microgliocytes contacted with the processes of astrocytes and were localized in the zone of distribution of numerous granules of unknown origin (Figure 14B). In general, microglial cells in the cerebellum represented a sparse population of small, highly branched cells (Figure 14C).

## 3. Discussion

The conducted studies allowed us, for the first time, to identify the features of the ultrastructural cerebellum organization in juvenile chum salmon, *O. keta*, within the framework of the traditional structural plan of the bony fish cerebellum. Three anatomical zones were identified in the juvenile chum salmon cerebellum: the CC (*corpus cerebellum*), the VC (*valvula cerebelli*) and the GrEm (*eminentia granularis*), which included the ML, GrL, and GL [35]. The conducted studies showed that the cerebellum of juvenile chum salmon has a general structural plan characteristic of other bony fishes [35,36]. An ultrastructural analysis of all cerebellar layers revealed the features of the neuronal, synaptic and glial cytoarchitecture, which determine the high mobility, neurogenic activity and behavioral characteristics of juvenile chum salmon (Table 1). One of the noteworthy observations was the detection of a variety of macro- and microglia in the cerebellum, which distinguishes juvenile chum salmon from other bony fishes and supports previous findings on the mesencephalic tegmentum [37]. As a result of the ultrastructural study, differences and similarities in the organization of heterogeneous interneuron groups of the cerebellum were found; details of the microcytosculpture of liana-like afferents and glial and non-glial precursors of adult types (aNSPCs) were determined; and astrocytic endocytosis in the area of synaptic contacts of parallel fibers and dendrites of the PC were described in detail.

Studies on fish have shown that, histologically, the cerebellar cortex of teleosts exhibits a much less orderly layered structure compared to the more clearly organized trilayered cortex of tetrapods [31]. The cerebellar cortex of mammals consists of three layers, including an outer ML with a small number of somatic neurons, a monolayer of Purkinje cells, and a deep GrL consisting mainly of small GrCs [28]. This distinct layering has been found to be generally characteristic of non-mammalian vertebrates, with the exception of fish and cyclostomes [3]. These characteristic zones of the cerebellar cortex and their corresponding cell types are significantly more diverse in juvenile chum salmon, which is consistent with data for other fish species [5,6]. In particular, in juvenile chum salmon, PCs can be found in the GrL, and GrCs are located laterally to the GL.

Previous studies have shown that teleost fishes lack basket cells [38]. In our studies on chum salmon, we did not find these cells as well. Therefore, negative feedback loops in the cerebellum are created only by GCs and SCs. The EDCs, found in juvenile chum salmon, apparently replace the deep cerebellar nuclei in terrestrial vertebrates, which is consistent with data on other fishes [39,40].

### 3.1. Features of the Histological Structure and Immunohistochemical Labeling of the DMZ in the Cerebellum of Juvenile Chum Salmon O. keta

It is known that in adult fish, as in mammals and birds, the cerebellum is the largest compared to that in other vertebrates [31]. Another feature of the fish cerebellum is its high structural diversity, being one of the forms of plasticity [36,41]. The histological organization of the DMZ in juvenile chum salmon is characterized by heterogeneity of the cellular composition and includes a few elongated, light-colored cells of the neuroepithelial type, located along the periphery of the DMZ, and numerous dark-colored stellate-like cells, which are aNSPCs and occupy the central part of the DMZ.

Immunohistochemical (IHC) studies of the DMZ features in juvenile masu salmon, *O. masou*, showed the presence of PCNA+ and bromodeoxyuridine+ (BrdU+) cells, which indicates the proliferative activity of the cells in the intact cerebellum and a significant increase in the number of such cells after injury [15]. Small, densely stained, stellate-shaped cells were found in the dorsal and ventrolateral regions of the DMZ of juvenile chum salmon, *O. keta* (Table 1). The ultrastructural organization of the DMZ, the main neurogenic region of the juvenile chum salmon cerebellum, was dominated by cells with dense cytoplasm, stellate morphology, and a large, irregularly shaped nucleus, which determines the glial phenotype of the aNSPCs according to [30]. In contrast to the *Apteronorus leptorhynchus*, characterized by the dominance of embryonic-type precursors (NECs) in the cerebellum, the majority of cells in the DMZ in juvenile chum salmon were represented by aNSPCs, which is consistent with data on *Danio rerio* [42]. IHC labeling of glutamine synthetase in the juvenile chum salmon cerebellum showed the presence of immunopositive cells in the DMZ [17], which supports the current TEM data on the presence of glial aNSPCs. The results of studies on masu salmon showed the presence of vimentin+ [16], GFAP+ cells [15] in the DMZ, which are glial markers and identify adult-type precursors in this region.

The discovery of non-glial precursors in the DMZ of juvenile chum salmon, characterized by rounded shapes, a large, light, round nucleus, and light cytoplasm, was a remarkable finding of this work. Previous studies on masu salmon demonstrated the presence of nestin+ and PCNA+ cells in the cerebellum [16]. Similar immunolabeling results were obtained for the trout cerebellum [14]. Although the functions of nestin in the central nervous system are not yet fully understood, it is clear that nestin is an effective marker for studying aNSPCs behavior and nervous system development [43]. The presence of nestin+ cells in the DMZ and neurogenic zones of the granular eminences (GrEm) in closely related salmonid species, as well as the localization of superficial neurogenic clusters identified in juvenile chum salmon, indicates the persistence of a neotenic structure in the juvenile salmon cerebellum. The phenomenon of embryonization of the salmon brain is confirmed by the detection of non-glial aNSPCs in the juvenile chum salmon cerebellum using TEM and SEM. The results of TEM and SEM analyses on juvenile chum salmon, along with previously identified nestin+ and PCNA+ cells [13,15], suggest that the identified adult-type progenitors represent a population of non-glial progenitors that retain neurogenic activity and contribute to homeostatic growth of the cerebellum in juvenile chum salmon.

### 3.2. Molecular Layer

#### 3.2.1. Features of the Synaptic Structure of the SCs

Several types of structures, including parallel fibers (pfs), PC and EDC dendrites, and SCs, were found in the ML of the juvenile chum salmon cerebellum. SCs are a relatively small population of cells with multipolar morphology, a small stellate soma, and a large nucleus. According to previous investigations, fish SCs are cells with GABAergic neurotransmission, along with other cerebellar interneurons such as PCs and CGs [44]. Imaging studies in mice have shown that pfs provide input to SCs which extend axons parallel to pfs and form inhibitory synapses with PC dendrites to modulate pf input [29]. This suggests that the projections of pfs and SCs play a central role in the integration and processing of incoming information in the mammalian cerebellum [45]. In studies on juvenile chum salmon, synaptic contacts between SCs and pfs were not detected.

According to ultrastructural data, numerous synaptic structures are formed in the ML of juvenile chum salmon, including one of the largest and most significant synapses in the cerebellum, between pfs and PC dendrites, providing excitatory input to the PCs. For the first time, we identified morphologically heterogeneous synaptic terminals in this region of the ML in juvenile chum salmon (Figure 4B,C). It is known that PC dendrites extend to the ML, while their somata are located in the GL. Most of the excitatory-type pf synaptic terminals of juvenile chum salmon converge on the dendritic bushes of the PCs, which largely resemble the analogous structure in higher vertebrates and zebrafish, *D. rerio* [29].

The excitatory terminals of the cfs were identified on the PC bodies of juvenile chum salmon (Figure 9E); synapses between cf and PC are the second most important synaptic structures in the cerebellum. SEM studies of cfs of juvenile chum salmon showed the presence of numerous terminal thickenings and varicose microcytosculpture (Figure 7A,B), which most likely correspond to areas of synaptic contacts.

The neural activity of climbing fibers suppresses synaptic transmission from GrC axons through a mechanism known as long-term depression, which is required for motor learning [46,47,48]. In fish, cfs send their axons either to neighboring PCs or to EDCs [1,29], while EDCs send efferent axons to other brain areas [22], forming synapses between PCs [49,50]. Some fish PCs also form connections with other PCs. It should be noted that PCs in the lateral CC send direct efferent signals to the vestibular nuclei (cerebello-vestibular tract), which may play a role in early vestibular processing [51]. Thus, as in the case of GCs, there is a small proportion of PCs that are not involved in intracerebellar connections but represent an efferent cell population.

We observed similar patterns of PCs and EDCs localization, but found no direct synaptic connections between these cells in the juvenile chum salmon cerebellum.

#### 3.2.2. Adult-Type Neural Stem Progenitor Cells (aNSPCs) in the Molecular (ML) and Granular Layers (GrL) of the Cerebellum

Compelling evidence for ongoing adult cerebellar neurogenesis has been shown in cypriniformes: zebrafish [52], the goldfish *Carassius auratus* [53], cichlids *Astatotilapia burtoni* [54], the killifish *Nothobranhius furzeri* [55], the *Oryzias latipes* [56], the electric brown ghost fish *Apteronotus leptorhynchus* [57], and salmonids: the trout *O. mykiss* [14], the chum salmon *O. keta* [17], and the masu salmon *O. masou* [15,16]. In mammals, evidence of spontaneous adult neurogenesis has been documented for the rabbit cerebellum [58].

As regards the homeostatic neurogenesis in salmonids, evolutionarily ancient representatives of teleosts, the question as to why neurogenesis continues after embryonic or juvenile stages of development, allowing Pacific salmon to attain substantial body sizes, still remains unanswered [14,59,60]. In the ML of the juvenile chum salmon cerebellum, a type of small cells was identified that represented the population of aNSPCs with a large, irregularly shaped nucleus and a narrow rim of cytoplasm. Among the aNSPCs in the ML of the cerebellum, both polarized cells with a large, irregularly shaped dark nucleus (type III) and cells with a light, irregularly shaped nucleus containing heterochromatin (type IV) were identified. Such cells were found in the parenchyma of the ML, approximately in the upper third, and were also identified by SEM on the dorsal and lateral surfaces of the cerebellar body (Figure 7A–C). These cells were also involved in homeostatic neurogenesis, since they were prone to proliferative activity and the formation of surface clusters (Figure 7D). In many fish species, structure-specific neurogenesis can be associated with the growth mode [36].

Many teleosts exhibit the phenomenon of indefinite cerebellar growth, in which cells continue to be added throughout the body, including the CNS, as the fish grows [61]. This has been demonstrated, e.g., in goldfish, which continue to grow for many years. In the brown ghost fish, considered as a model of mild aging, which also exhibits indefinite growth, up to 75% of all mitotically active cells are localized in the cerebellum [57]. In *A. leptorhynchus*, proliferative activity is observed in the DMZ along the CC midline, as well as in the VC and in the GrEm. Similarly, in zebrafish, the cerebellum grows proportionally more than other brain regions at the juvenile stage (30–90 days post-fertilization), and the CC containing GrCs exhibits significant growth throughout life [30]. However, studies on zebrafish have shown that this species exhibits determinate growth [62], consistent with the growth limitations observed in amniotes. Nevertheless, evidence from various fish species, including juvenile chum salmon, raises the question of why constitutive NSPCs proliferation is required in the cerebellum throughout life.

The results of ultrastructural analysis confirm previous IHC studies on juvenile masu salmon, *O. masou*, with vimentin labeling [16] and on juvenile chum salmon, *O. keta*, with vimentin and glutamine synthetase immunolabeling [8,17]. According to the results of these studies, Vim+ and GS+ populations of superficial small cells are present in juvenile salmonids, and their number increases dramatically after exposure to acute traumatic injury [16,17]. We suggest that type III aNSPCs in intact juvenile chum salmon may be in a quiescent state and are not involved in homeostatic proliferation. In contrast, type IV aNSPCs may function as homeostatic precursors involved in constitutive growth of the cerebellum. Our data are consistent with the results of a study in zebrafish, in which resident neural stem cells in the postnatal cerebellum, known as “stem cell niches”, continue to generate newborn neurons [63].

However, even in basic teleost fish models such as zebrafish, the extent of structural brain growth and neural stem cell activity is dramatically reduced in older individuals [36,63], suggesting that continuous brain growth is not unlimited. In zebrafish studies, more than 16 major neurogenic domains have been shown to exhibit constitutive neural stem cell proliferation [64,65]. However, as Zupanc reported, more than 100 neurogenic regions can be detected [66]. In contrast, adult mammals have two main neural stem cell niches, which are limited to the subventricular zone of the forebrain and the subgranular zone of the hippocampus [67].

Studies by Kaslin and Brand have shown that the proliferative activity of stem and progenitor populations of the zebrafish cerebellum decreases after the juvenile period (up to 3 months). However, in adulthood, progenitors originating from the superior rhombic lip continue to produce granular cells [30]. In zebrafish, no new PCs or EDCs are formed after the juvenile stage. This reduced degree of postembryonic neurogenesis in the cerebellum coincides with the onset of a plateau in the growth of zebrafish compared to its close relative, the giant zebrafish [62].

The results of studies on 1.5-year-old juvenile chum salmon showed that in the region of the neurogenic zones of *torus semicircularis*, as well as *valvula cerebelli*, a large number of aNSPCs in a state of constitutive proliferation are retained, with this number significantly increasing after injury [36]. The detection of apoptotic bodies and heterogeneous groups of astrocytes and microgliocytes in the cerebellum (in the present study) and in the tegmentum of juvenile chum salmon [37] indicates homeostatic apoptosis processes. These findings strongly suggest, that newly formed cells contribute to maintaining the brain homeostasis by replacing the population of cells that have died. Comparative studies on models of both determinate (zebrafish, *D. rerio*) and indeterminate growth (chum salmon, *O. keta*) are of certain interest to elucidate whether aNSPC populations of the cerebellum are capable of characteristic responses to environmental influences. In particular, further studies on juvenile chum salmon may address responses to sensory or motor stimuli that have been found in other stem cell niches of the adult zebrafish brain [65,68].

#### 3.2.3. Non-Glial Progenitors in the DMZ and GrL and Neoteny

Understanding the cellular processes occurring at later stages of ontogenetic development is important for addressing age-related changes. The identification of a large population of non-glial aNSPCs in the DMZ and GrL of juvenile chum salmon, along with stromal cell clusters on the surface of the cerebellar ML, suggests the presence of a neurogenic and hematopoietic program in its brain, which is active primarily at embryonic stages in other vertebrate species [66,67]. The cerebellum is only one of the regions of the salmon brain where similar cell types were identified by SEM and TEM in the present study. Previously, using IHC methods, we identified similar neurogenic niches in various parts of the brain of adult trout *O. mykiss* [14,60], masu salmon *O. masou* [15,16], and chum salmon *O. keta* [17]. The presence of constitutive neurogenic clusters with proliferating cells in juvenile and adult Pacific salmon is a neotenic characteristic of the brain. It is known that in all vertebrate species, the embryonic period is characterized by a higher rate of cell proliferation compared to the adult phase [69]. Furthermore, in some species such as salamanders and frogs, an extended larval period results in increased body size [70,71,72]. This supports the hypothesis that the observed neotenic status in the juvenile chum salmon cerebellum may be a necessary adaptation for maintaining very high levels of metabolism and cell proliferation during homeostatic growth.

Our data confirm the results of studies on the killifish *Nothobranchius furzeri*, which is one of the recognized shortest-lived vertebrates capable of breeding in captivity, with a lifespan of 3 to 8 months [56,73]. This species undergoes a rapid natural aging process as an adaptive response to the environment. Despite its short lifespan compared to other vertebrates, *N. furzeri* displays many physiological features of aging found in other vertebrates [73]. Recent studies in *N. furzeri* have shown that non-glial progenitor cells are among the aNSPC populations that change most significantly between young and old animals [56,73]. It was found that embryonic gene expression in the adult brain is a common feature in *N. furzeri* and that several cell populations in the adult *N. furzeri* brain exhibit high levels of similarity to zebrafish embryonic populations. These data indicate that adult *N. furzeri* maintain a neotenic gene expression state in the brain, which may support their characteristic high rates of proliferation and metabolism.

The results of ultrastructural studies of the juvenile chum salmon cerebellum show the persistence of neotenic structure at the level of organization of neurogenic and hematogenous cell clusters located in different regions of the cerebellum. Such cell types were identified as components of superficial neurogenic clusters in the ML, often containing erythrocytes, as well as in the GrL. A noteworthy feature of these structures is their potential similarity to embryonal clusters of zebrafish, as well as their involvement in neotenic growth of juvenile chum salmon, contributing to a significant increase in body size. Our findings are consistent with the results of studies on *N. furzeri*, which demonstrated neotenic features at the level of gene expression over a longer period and even persisted for some programs throughout life (unpublished data). This hypothesis was supported by direct comparison of individual cell clusters of *N. furzeri* with zebrafish embryo samples which revealed a high degree of correlation.

Ultrastructural analysis of non-glial progenitors in the juvenile chum salmon cerebellum provides insight into the evolutionary mechanisms that cells must acquire to adapt to the species’ environment. Juvenile chum salmon is a good model for studying homeostatic growth and responses to traumatic injury; however, extrapolation of these findings to other vertebrate species, such as mammals, should be approached with caution. Nevertheless, the neotenic status observed in juvenile chum salmon may be part of the species’ adaptation to its demanding ecological niche and may help identify promising cellular and molecular targets for regenerative or anti-aging therapies in the future.

### 3.3. Ganglion Layer

In terms of branching, the dendritic tree of PCs in fish is more complex than in amphibians and reptiles but is never as extensive as in mammals [5]. In teleosts, the proximal, smooth part of the dendritic tree, which contains the receptive surface for ascending fibers, does not penetrate the molecular layer as it does in mammals. This has been best demonstrated for mormyrids, but is suggested to be common among many teleost species [43].

There are several differences between the cerebellum of teleosts and the cerebellum of mammals. The mammalian cerebellum lacks EDCs and contains deep cerebellar nuclei that play a functionally homologous role as they receive axons from PCs and send efferent axons outside the cerebellum [29]. In mammals, these intrinsic cerebellar nuclei are located ventrally, far from the PCs, whereas in teleosts, EDCs are located in close proximity to the PCs. Although both the mammalian cerebellar intrinsic nuclei and the teleost EDCs are projection neurons that send axons outside the cerebellum, it remains unclear whether they use the same molecular mechanisms for their function and development. They likely correspond to a modified version that arose through evolutionary divergence from the original or ancestral cerebellar nucleus.

The evolutionarily most ancient cartilaginous fishes have a well-defined cerebellar nucleus with subdivisions [74,75]. In the cerebellum of sturgeon, which belongs to the most ancient group of bony fishes, EDCs are grouped in three regions of the cerebellum and are less branched than in zebrafish [76]. In addition, the EDCs of the basal actinopterygian fish *Polypterus senegalus* are similar to those of teleosts, as well as to the deep nuclei of the mammalian cerebellum [77]. Thus, the EDCs in the most ancient bony fishes are assumed to correspond to an intermediate stage before the emergence of true EDCs in bony fishes [78].

As a result of our studies on juvenile chum salmon, we found that part of the PC population also contains a receptive surface for ascending fibers without penetrating the ML, which is consistent with data on *D. rerio* [29]. However, the PC population in juvenile chum salmon is quite heterogeneous: in particular, we found four types of morphologically distinct PCs, which is consistent with data from studies on zebrafish [30]. PCs in juvenile chum salmon, as in all groups of gnathostomes, are considered the main neurons of the cerebellum and represent its main information processing center. IHC studies on juvenile masu salmon showed that the PCs are parvalbumin-ip, GABA-ip, and CBS-ip [78]. Immunolocalization of GABA and the calcium-binding protein parvalbumin was detected in bodies and dendrites of the PCs, while the hydrogen sulfide synthesis enzyme was found to be localized in the fibers of the molecular layer [78].

In zebrafish, the PCs express Zebrin-II. However, in this neuronal population, instead of a zebra-like compartmentalization as in mammals, all PCs are Zebrin-II (or aldolase-C) immunoreactive [79], as in cartilaginous fish [80,81]. There are exceptions in some fish, in which some PCs have been reported to be Zebrin-II negative, but the PC population still does not exhibit layered segregation [1]. Regardless of Zebrin-II expression, there is evidence that zebrafish PCs represent a heterogeneous population of cells. Recent studies of the mature zebrafish cerebellum have shown that PCs are involved in both locomotor and non-locomotor behaviors [50,82].

The results of the present study showed that the GL in juvenile chum salmon is not strictly organized, and part of the PCs appears ectopic in the GrL. The GL cells are surrounded by extra- and infraganglionic plexuses, within which apoptotic bodies were identified. Previously, apoptotic patterns were identified in the tegmentum of juvenile chum salmon [37,83]. As known, apoptosis, which is a programmed type of cell death, does not provoke an inflammatory reaction in the fish CNS [84]. The outer membrane of the cell remains intact, and adjacent cells and tissues are not damaged [85]. We believe that the death of undifferentiated cells formed during homeostatic neurogenesis is a consequence of natural differentiation and elimination of cell forms containing defects.

As shown by studies on *Apteronotus leptorhynchus* [86], if undifferentiated cells do not have time to establish connections with fibers and/or other types of cells, caspases are activated in them and cells undergo apoptosis. A study of aneuploid cells lacking genes that support vital activity in the brain of *A. leptorhynchus* provided evidence that such cells are destroyed by apoptosis not only in the early stages of development, but also in the later stages of ontogenesis, when the consequences of inadequate gene expression appear [87].

The main morphological signs of apoptosis recorded from the cells of the infraganglionic plexus of juvenile chum salmon were the cell shrinkage, cytoplasmic condensation, membrane swelling, nuclear fragmentation, and formation of apoptotic bodies (Figure 9A). We suggest that in juvenile chum salmon, the latter are promptly and orderly eliminated by macrophages and resident microglia and are also found in the infraganglionic plexus region of the ganglion layer (Figure 9F and Figure 14A–C). Studies on the cerebellum of *A. leptorhynchus* confirm that one of the main functions of microglia and macrophages may be removal of remnants of cells that have undergone apoptosis at the site of injury [86]. It has been hypothesized that this programmed type of cell death is a major factor in the rapid and, sometimes, complete regeneration of nervous tissue [84]. Similar conclusions were made in a study of traumatic eye injury in adult trout, *O. mykiss* [88], and in a study of Wallerian degeneration of the optic tract in *Carassius auratus* [89]. In a study conducted at two days after spinal cord transection in zebrafish, along with myelin phagocytosis, a macrophage/microglia response was also observed caudally of the transection site using IHC and electron microscopy [90].

In the area of the infraganglionic plexus, we found patterns of small, sprouting, weakly myelinated or unmyelinated fibers, indicating homeostatic axonogenesis in the juvenile chum salmon cerebellum.

#### 3.3.1. Neuro–Glial Relationships in the Ganglion Layer of the Juvenile Chum Salmon Cerebellum

Microglia and astrocytes play important roles in the fish CNS, contributing to homeostasis, immune response, and maintenance of the blood–brain barrier and synaptic support [59,86]. Studies of various experimental models of glial communication have provided evidence that microglia and astrocytes influence and coordinate each other and their impact on the brain environment [91,92]. The detection of macroglia and microglia populations adjacent to the projection PCs in juvenile chum salmon using TEM was an important finding. The detection of astrocytes is very important for identifying neuro–glial relationships in the juvenile chum salmon cerebellum. These data are consistent with the results of studies of the chum salmon mesencephalic tegmentum where astrocytes and microgliocytes were also found [37]. The protoplasmic astrocytes we detected in the cerebellum formed contacts with the PCs.

Our observations contrast with the data on zebrafish, in which astrocytic glia have not been detected [44]. Along with astrocytes, we observed single ramified microgliocytes adjacent to astrocytes in the GL (Figure 14A–C). The detection of a population of astroglia and microglia in the GL of the juvenile chum salmon cerebellum significantly expands current understanding of the cellular heterogeneity and functionality of homeostasis in the juvenile cerebellum of this species.

#### 3.3.2. Astrocytes

Astrocytes are a subgroup of glial cells that help maintain and regulate neuronal function, including neurotransmitter cycling, metabolic support for neurons, and maintenance of the blood–brain barrier [93]. In humans, the proportion of astrocytes in the brain ranges from 17 to 61%, depending on the region [94]. According to our observations, in the juvenile chum salmon cerebellum, the PC-to-astrocyte ratio is 2:3.

Previous IHC studies on juvenile masu salmon, *O. masou*, showed the presence of GFAP+ cells and fibers in the DMZ [15]. Subsequent studies of vimentin and glutamine synthetase immunolocalization in the cerebellum of juvenile masu salmon *O. masou* [16], chum salmon *O. keta* [17], and trout *O. mykiss* [14] also confirmed the presence of an astrocytic population and radial glia, which contrasts with data on zebrafish [44]. Thus, the juvenile chum salmon cerebellum has a population of astrocytes expressing glial markers, largely resembling the human brain model [92,94]. Thus the question as to whether the morphological heterogeneity of astrocytes in the juvenile chum salmon cerebellum is combined with the existence of astrocytic phenotypes known for the mammalian brain—the neurotoxic phenotype (A1) and the neuroprotective phenotype (A2) is becoming increasingly relevant.

According to the results of transcriptome analysis, intermediate or divergent transcriptome states can exist and even coexist in the same brain [95]. In mammals, during neuroinflammation, astrocytes acquire a neurotoxic phenotype (A1) and additionally express GFAP [96], which loses its neuroprotective abilities, leading to neuronal death. This A1 phenotype has been identified in Alzheimer’s disease, Huntington’s disease, motor neuron disease, and Parkinson’s disease [97].

The finding of astrocytes in the juvenile chum salmon cerebellum is important in terms of neuroinflammatory processes. As a result of acute injury, the number of GFAP+, vimentin+, and GS+ cells in the juvenile salmon cerebellum increases [15,16,17]. However, the increased production of glutamine synthetase during injury in juvenile salmon leads to a decrease in the excitotoxic effects of glutamate by neutralizing it to glutamine [16,17,60]. The heterogeneous populations of protoplasmic astrocytes in the juvenile chum salmon cerebellum, identified using TEM, are consistent with the assumption of probable coexistence of different phenotypes under homeostatic growth conditions in juveniles.

It is known that changes in the morphology, function, and number of astrocytes, referred to as “astrogliosis”, are observed as a result of traumatic brain injury and in human neurodegenerative diseases [98]. However, many studies used GFAP to monitor astrocyte proliferation, suggesting that increased expression may be due to rather upregulation in individual cells than astrocyte proliferation [99], as observed in a study using the proliferating cell marker PCNA [100]. Furthermore, studies using astrocyte counting, via Nissl assays [101] or constitutive astrocyte markers in combination with GFAP [102], showed no difference in the number of cells between the brain affected by Alzheimer’s disease and the intact brain. In contrast, following cerebellar injury in juvenile masu and chum salmon, a significant increase in the number of PCNA, GFAP, Vim positive cells was observed [15,16]. Nevertheless, the different regenerative pathways involved in cerebellar repair in fish and mammals actualize new areas of interest in studying the biology and functional properties of cerebellar astrocytes in chum salmon.

The study of astrocyte properties in the juvenile salmon model is relevant for several reasons. First, unlike that in higher vertebrates, regeneration in the cerebellum of juvenile salmon is successful and is obviously associated with the existence of special phenotypes with increased expression of glutamine synthetase and other neuroprotective factors such as hydrogen sulfide and aromatase B [17]. Second, the unique properties of astrocytes in the juvenile salmon brain can clarify which combinations of molecular markers are associated with successful reparation. Thus, in humans, there are two different subtypes of astrocytes, which are located either in layer I or in layer VI and project to other cortical layers [103]. These interlaminar astrocytes have high expression of CD44, GFAP, and S100B (calcium-binding protein B S100), but low expression of glutamate processing markers, excitatory amino acid transporters EAAT1 and EAAT2, and glutamate synthetase [104]. Similar transcriptional properties have also been found in the “fibrous astrocytes” of the white matter. Since the majority of astrocytic glial cells in the fish brain are represented by radial glia and often retain their radial identity throughout life [105,106], the protoplasmic-type astrocytes found in the juvenile chum salmon cerebellum are the final stage of astrogliogenesis. We suggest that these cells support metabolism in projection interneurons of the ganglion and granular layers and influence neural connections by forming tight contacts with Purkinje cells (Figure 5C, D). We also assume that the cells with dense, dark-stained cytoplasm and several processes found in the cerebellum, as those previously found in the tegmentum [37] of juvenile chum salmon, are astrocytes that have separated from the ventricle and have a typical astrocytic phenotype. The differences consist in the structural features of the nuclei of these cells. While the tegmental astrocytes clearly visualized end-feet and a single nucleus of irregular shape [37], in the astrocytes identified in the cerebellum, end-feet were rarely detected, and binucleation was also observed.

Another property of cerebellar astrocytes in juvenile chum salmon is their phagocytic ability. In the area of synaptic contacts, vesicles with secretory granules, possibly neurotransmitters, were found that undergoing astrocytic endocytosis by forming elongated processes resembling villi and capturing single vesicles (Figure 13A,B). These astrocytes were relatively large, with an irregularly shaped nucleus. They contain glutamine synthetase, which converts toxic glutamate to neutral glutamine [59,107]. We assume that the capture of vesicles with glutamate by astrocytes promotes the metabolism of glutamate and its conversion to a neutral form to eliminate excitotoxicity. It is known that astrocytes are involved in synaptic transmission through the regulated release of glutamate, GABA, and D-serine [93]. It was previously reported that GABAergic neurons are localized in the juvenile cerebellum of a closely related species, masu salmon *Oncorhynchus masou* [78]. The release of gliotransmitters, accompanied by an increase in the Ca^2+^ current in astrocytes, occurs in response to changes in the synaptic activity of neurons [108] and can modulate the activity of large GABAergic interneurons in the juvenile chum salmon cerebellum.

In addition to directly influencing synaptic activity through the release of gliotransmitters, astrocytes have the potential to exert potent and long-lasting effects on synaptic function through the release of growth factors and cytokines [109]. Astrocytes are also sources of neuroactive steroids (neurosteroids), including estradiol, progesterone, and aromatase, which can exert synaptic effects on GABA receptors [17,110,111]. Studies on the juvenile chum salmon tegmentum have shown increased aromatase B expression in response to traumatic injury [37]. The different types of astrocyte-neuron contacts identified in the cerebellum of juvenile chum salmon (Figure 12B–D) allow the activation of astrocytes through neurotransmission, since astrocytes are known to express a variety of glutamate receptors [111]. Upon activation of astrocytes, signaling pathways involving extracellular signal-regulated kinase (ERK) and c-Jun N-terminal kinase (JNK) are triggered [112].

#### 3.3.3. Microglia

Microglia are an important cell population involved in homeostatic maintenance of the fish brain. These cells exhibit variable proliferative activity and, under normal conditions, maintain a ramified morphological identity consistent with the resting state, similarly to the mammalian brain. Studies in zebrafish have shown that microglia are capable of transforming into an amoeboid, activated form under pathological conditions [113]. When involved in CNS defense reactions, microglia in fish play an important role in synaptic degeneration [114], regulation of neuronal components [115,116], clearance of apoptotic cells [117], and in CNS angiogenesis and vascular maintenance [118].

The results of ultrastructural studies on juvenile chum salmon showed the presence of ramified (resting) microglia (Figure 14A–C). This type of cell was not found in the juvenile chum salmon tegmentum [37]. Microgliocytes are resident macrophages of the brain and primary immune cells of the CNS, rapidly responding to injury, infection, and inflammation [91]. Resident mammalian microgliocytes are believed to continuously interact with their environment throughout adult life, facilitating synaptic communication and maintaining cerebral homeostasis by constantly surveying the surrounding parenchyma with their finger-like processes [119]. They have recently been identified as a multifunctional type of “housekeeping” cell [120]. However, the exact function of microglia is still unclear.

Microglia are heterogeneous CNS cells that maintain homeostasis by surveying the environment with branched processes. Upon activation by signals like ATP, their processes retract and thicken, adopting an activated phenotype [91]. When activated, the ramified processes of microgliocytes retract and thicken, causing the cell to adopt an activated phenotype. Among fish, the best studied, including at the ultrastructural level, are the microglial cells of the optic nerve of *C. auratus* [121]. Activated microglial cells (macrophages) in the juvenile chum salmon cerebellum, identified as part of microvessels, are similar to those in the tegmentum [37] and also resemble mammalian microglia, with a heterochromatic nucleus and a high content of extended RER [91].

Various methods have been used to identify microglial cells in teleost fish [90,122,123], among which lectin histochemistry has proven to be an effective tool for detecting microglia. Lectins from *Lycopersicon esculentum* have an affinity for poly-N-acetyllactosamine sugar residues and have been shown to identify amoeboid and branched microglial cells in the neonatal and adult rat brain [124], microglial cells in normal and injured fish retina and optic nerve [125], and a microglial cluster in the pufferfish *Diodon holacanthus* in supramedullary neurons [126].

Studies on the juvenile chum salmon cerebellum showed that microglia are able to dynamically interact with synapses of the infraganglionic plexus and ganglion layer, changing their structure and function and being involved in the maintenance of cerebellar homeostasis. Ultrastructural data show that microglia interact with astrocytes (Figure 14B,C). These interactions may produce soluble factors in extracellular vesicles in the GL and GrL, supporting local neuronal integrity. On the other hand, soluble factors contained in microglia appear to regulate neurogenic differentiation of aNSPCs, which is consistent with previous findings [118,127]. In particular, IL-6 and LIF molecules released by activated microgliocytes have been shown to promote glial differentiation of aNSPCs, highlighting the importance of cell–cell interactions between glial cells and NSPCs [128].

Studies of microglial behavior in fish are becoming increasingly common, in particular those focusing on the microglial response to injury through phagocytosis [129,130]. In ultrastructural studies on the juvenile chum salmon cerebellum, we found patterns of phagocytosis, as well as apoptotic bodies, indicating the presence of these processes during the homeostatic growth mode of juvenile *O. keta*. The phagocytic behavior of immune cells and astrocytes is potentially important for establishing new connections in the brain during homeostatic growth, and also plays a significant role after injury, since degenerating connections must be removed first before new connections can be formed [131].

Data from zebrafish trauma studies show the development of a microglial inflammatory response characterized by morphological modification and accumulation of leukocytes at the injury site [132,133]. During the development of the inflammatory response in the zebrafish telencephalon, microglial cells migrate away from the injury site following the inactivation of sigma-1 receptors [134]. The latter are intracellular proteins in neurons and glia known to play a role in neurodegeneration. Studies have examined the recruitment of both resident microglia and peripheral macrophages to brain injury and their distinct roles during the neuronal cell death response [135,136]. Further studies on juvenile chum salmon will help clarify the involvement of microglia in anti-inflammatory processes in traumatic brain injury.

## 4. Materials and Methods

### 4.1. Experimental Animals

The study was conducted on 15 juvenile (aged 1 year and 2 months) Pacific chum salmon, *Oncorhynchus keta*, with a body length of 20.0 ± 1.6 cm and a body weight of 42.5 ± 3.8 g. The animals were provided by the Ryazanovka Experimental Fish Hatchery (Ryazanovka, Russia) in 2024. Most of the fish used in this study were males. The chum salmon specimens were kept in a tank with aerated freshwater at a temperature of 13–14 °C and fed once a day. The light-dark cycle was 14/10 h. The concentration of dissolved oxygen in the water was 7–10 mg/dm^3^, which corresponds to normal saturation. All experimental manipulations with the animals were carried out in accordance with the regulations of the charter of the A.V. Zhirmunsky National Scientific Center of Marine Biology, Far Eastern Branch of the Russian Academy of Sciences (NSCMB FEB RAS) and the Ethics Committee regulating the humane treatment of experimental animals (approval No. 9 dated 19 May 2025, at meeting of the Biomedical Ethics Committee of the NSCMB FEB RAS). The specimens were divided into three groups. The first group consisted of intact specimens and were used to study the histological structure of the brain on semi-thin sections (*n* = 5). The experimental groups consisted of specimens selected for transmission electron microscopy (TEM) (group 2, *n* = 5) and intact specimens for scanning electron microscopy (SEM) (group 3, *n* = 5). Specimens were anesthetized with 0.1% tricaine methanesulfonate (MS222) (Sigma, St. Louis, MO, USA, Cat. # WXBC9102V) for 10–15 min. Animals were then euthanized by rapid decapitation.

### 4.2. Transmission Electron Microscopy

Transmission electron microscopy was used to study the ultrastructural profile of neurons and glia within the CC. For histological control, semi-thin sections of the CC and GrEm in the frontal plane were examined. Semi-thin (500-μm-thick) sections were mounted on glass slides at the corresponding rostro-caudal level of the cerebellum. The sections were stained with 1% methylene blue in 1% aqueous sodium tetraborate. After fixation of the mesencephalon in 2.5% glutaraldehyde in 0.2 M cacodylate buffer (pH 7.4) at 4 °C overnight, the samples were washed and then postfixed for 2 h with 1% OsO_4_ in 0.2 M cacodylate buffer. The tissues were then washed and dehydrated in an ascending ethanol series and then impregnated with 100% ethanol and LR-White resin. The following day, the tissues were impregnated with fresh LR-White resin twice over a 6 h period and then embedded in gelatin capsules and polymerized at 40 °C. Using a Leica UC7 ultramicrotome (Wetzlar, Germany), 500-μm-thick semithin sections were cut onto glass slides, and 50-60 nm thick ultrathin sections were cut onto formvar-coated copper grids at the corresponding rostro-caudal level of the cerebellum. Semithin sections were stained with 1% methylene blue in 1% aqueous sodium tetraborate. Ultrathin sections were stained with 2% aqueous uranyl acetate and lead citrate. Visualization was performed using a Zeiss Sigma 300 VP transmission electron microscope (Karl Zeiss, Cambridge, UK) and the AMT Image Capture Engine software (version 5.44.599). For analysis of the morphology of cells present in the niches of the ML and GrL of the CC and GrEm, they were compared with previously published seven ultrastructural profiles (Types IIa, Type IIb, Type III, Type IVa, Type IVb, Type V, and Type VI), described in the work “Neurogenic niches of the forebrain” [3,24] and «The cellular composition of neurogenic periventricular zones in the adult zebrafish forebrain» [34].

### 4.3. Cell Counting and Visualization

Visualization of semithin sections of the CC, VC, and GrEm was performed using an AxioImager Z2 microscope (Carl Zeiss, Göttingen, Germany). To study the ultrastructural organization of the cerebellum, low-magnification (2–4 K) TEM images were acquired at rostro-caudal levels. These images were used to define the topographic regions of the cerebellum: ML, DMZ, GL, infra- and extraganglionic plexuses, and GrL for subsequent cell type analysis. For morphological analysis of different cell types in different cerebellar layers (ML, GL, GrL) and DMZ, 5 to 15 cells of each type with morphologically unique ultrastructural features were selected. If only a few cells of a given morphology were observed within the DMZ, all identified cells were analyzed. The following features were studied to characterize the cell types: outline, color, chromatin organization, number of nucleoli, the greater and lesser diameters of the nucleus; percentage, color, presence of mitochondria, cilia, microvilli, vacuoles, lipid droplets, dense bodies in the cytoplasm; localization of cell types; and cell–cell contacts. Using these criteria, the morphological profile and frequency of each cell type were determined within each cerebellar layer (ML, GrL, GL) as well as within the DMZ and expressed as a percentage of all cells examined. Based on the identified morphological features, we developed a model representing the cerebellar cellular organization and a classification scheme for the different cell types constituting the different niches.

### 4.4. Scanning Electron Microscopy

Five brain samples of chum salmon specimens were fixed whole in 2.5% glutaraldehyde solution prepared in 0.2 M cacodylate buffer (pH 7.4) without adding NaCl for 24–48 h at 2–4 °C. The samples were then washed in 0.1 M cacodylate buffer for 20 min using a Mini-Shaker PSU-2T shaker (Biosan, Riga, Latvia). In the next step, the samples were dehydrated in alcohols of increasing concentration, gradually transitioning to pure acetone [137]. For this, the samples were first placed in a 7% alcohol solution for 10 min and then sequentially in 15, 30, 50, 70, 80, 90, and 96% alcohol solutions. Then the samples were placed for 10 min in a mixture of alcohol and acetone in ratios of 3:1, 1:1, and 1:3, gradually transitioning and finally to pure acetone. After that, the samples were dried by the critical point drying method, using the CPD 030 critical point dryer BAL-TEC (Pfeffikon, Switzerland). The dried samples of the cerebellum were mounted on aluminum stabs, pre-coated with carbon tape, which had been previously glued. Then the samples were sprayed with chromium, using a Q150T ES (London, UK) vacuum sputter coater for thin membrane coating. Then the structural features of the cerebellum were studied on a Sigma 300 VP scanning electron microscope (Carl Zeiss). Various parameters of the brain were measured using the Smartiff software. All obtained micrographs were processed using the Adobe Photoshop CS6 software beta version (22 March 2012) for Microsoft Windows.

## 5. Conclusions

The conducted studies allow a fresh look at the ultrastructural organization of the cerebellum of juvenile chum salmon and help fill the existing gap in ultrastructural studies of the cerebellum of salmonids. This is especially true for the interneuron composition of the juvenile cerebellum in this fish species, where the following cell types were identified using TEM: stellate cells in the ML, projection Purkinje cells, and EDCs in the GL. In the GrL, we found large GCs, as well as GrCs. Basket neurons were not identified in the present study, but the presence of large, elongated cells with bipolar morphology found in semi-thin sections does not rule out the existence of interneurons resembling Lugaro cells of higher vertebrates.

Of great interest is the synaptic structure of the molecular layer. In juvenile chum salmon, synaptic contacts of the electrotonic and chemical types were identified using TEM, which are important links in interneuronal communications. Ultrastructural data indicate that numerous synaptic structures are formed in the ML of juvenile chum salmon, including one of the largest and most significant synapses in the cerebellum, between parallel fibers and Purkinje cell dendrites, providing excitatory input to the PCs. Most excitatory pf synaptic terminals in juvenile chum salmon converge onto PC dendritic bushes, which largely resemble the similar structures in higher vertebrates and zebrafish.

TEM studies of neuro–glial relationships revealed the presence of a heterogeneous population of astrocytes and microglia in the juvenile chum salmon cerebellum. The detection of protoplasmic-type astrocytes in the juvenile cerebellum of this species contrasts with data reported for zebrafish. Homeostatic growth of the juvenile chum salmon cerebellum may correspond to an indeterminate growth model and be mediated by the presence of aNSPCs. The presence of adult glial and non-glial aNSPCs in the juvenile chum salmon cerebellum was demonstrated by TEM and SEM. The detection of an extensive population of non-glial aNSPCs in the DMZ and GrL of juvenile chum salmon, as well as stromal cell clusters on the surface of the cerebellar molecular layer, suggests the functioning of neurogenic and hematopoietic programs in the juvenile chum salmon brain, which are active mainly at embryonic stages in other vertebrate species. The phenomenon of embryonization in the juvenile chum salmon cerebellum is determined by the presence of non-glial aNSPCs, contributing to homeostatic growth. The neotenic status observed in the juvenile chum salmon cerebellum may be part of the species’ adaptation to a changing ecological niche and may help identify promising cellular and molecular targets for regenerative or anti-aging therapies in the future.

## Figures and Tables

**Figure 1 ijms-26-11123-f001:**
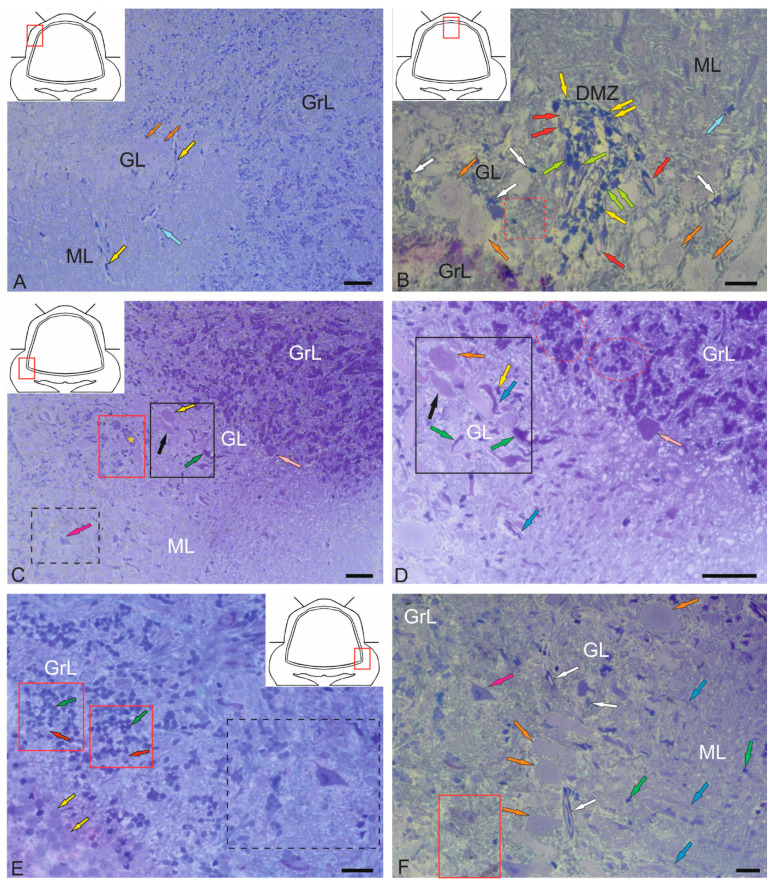
Semi-thin sections (stained with methylene blue) demonstrating the structural organization of the cerebellar body in juvenile chum salmon, *Oncorhynchus keta*. (**A**)—Dorsolateral region of the CC containing the molecular (ML), ganglion (GL) and granular (GrL) layers; Purkinje cells (PCs) are indicated by orange arrows; stellate cells (SCs), by blue arrow; vessels, by yellow arrows. (**B**)—Morphological structure of the dorsal matrix zone (DMZ): neuroepithelial cells are indicated by red arrows; small adult-type precursors, by yellow arrows; larger adult-type precursors, by green arrows; Purkinje cells (PCs), by orange arrows; astrocytes by white arrows; stellate cells in the ML, by blue arrows; patterns of sprouting fibers are outlined by red dotted square. (**C**)—Basolateral part of the CC region: Golgi cell (GC) is indicated by pink arrow; ectopic eurydendroid cell (EDC, magenta arrow) is shown in the thickened ML in the black dotted rectangle; PC is indicated by yellow asterisk in the red rectangle; ganglion cell cluster is outlined by black rectangle: piriform Purkinje cell (yellow arrow), bipolar eurydendroid cell (red arrow), and astrocyte (green arrow), bipolar cells (black arrow). (**D**)—Enlarged fragment in the black rectangle in (**C**): Bergmann glia is indicated by blue arrow; granule cell clusters (GrCs) are outlined by red dotted ovals; other designations are as in Figure (**C**). (**E**)—Basolateral part of GrL: heterogeneous GrL cell clusters are outlined by red squares; adult-type progenitors (aNSPCs) are indicated by green arrows; GrCs, by red arrows; non-glial progenitors, by yellow arrows. (**F**)—Basal region of ML: patterns of sprouting unmyelinated fibers are outlined by red rectangle; microvessels are indicated by white arrows; Bergmann glia, by blue arrows; PC, by orange arrows; astrocytes, by green arrows; EDC, by crimson arrow. Scale bars: (**A**,**C**,**D**)—50 µm; (**B**,**E**,**F**)—20 µm.

**Figure 2 ijms-26-11123-f002:**
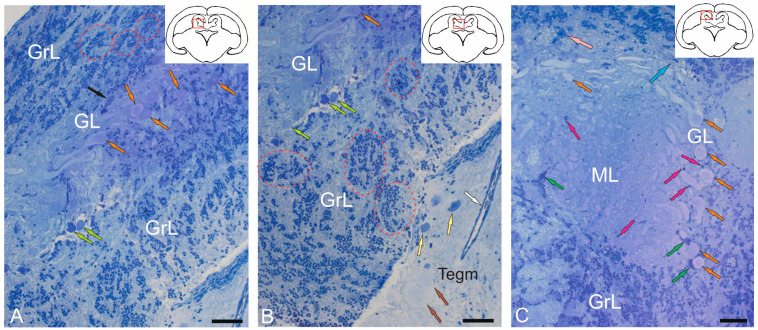
Semi-thin sections (stained with methylene blue) demonstrating the structural organization of the *valvula cerebelli* in juvenile chum salmon, *Oncorhynchus keta*. (**A**)—Rostro-lateral part of the VC: PCs are indicated by orange arrows; EDCs, by light green arrows; neurons with bipolar morphology, by black arrows. (**B**)—Medial part of the VC: clusters of heterogeneous cell types are outlined by red dotted ovals; neurons of the dorso-medial VC, by yellow arrows; microvessels, by white arrows; other designations as in (**A**). (**C**)—Dorso-medial part of the VC: PCs are indicated by orange arrows; large protoplasmic astrocytes, by green arrows; Bergmann glia, by blue arrow; astrocytes, by crimson arrows; cells of the supraganglionic region, by pink arrow. Scale bar: 50 µm.

**Figure 3 ijms-26-11123-f003:**
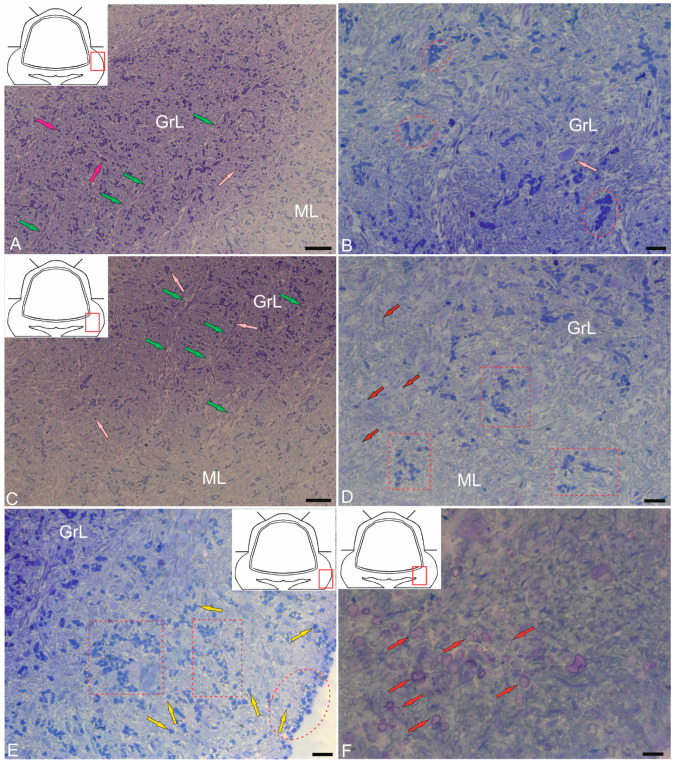
Semi-thin sections (stained with methylene blue) showing the structural organization of granular eminences (GrEm) in the cerebellum of juvenile chum salmon, *Oncorhynchus keta*. (**A**)—Morphological structure of GrEm containing molecular (ML) and granular (GrL) layers: fibers are indicated by green arrows; astrocytes, by crimson arrows; neurons, by pink arrows. (**B**)—A at higher magnification: clusters of aNSPCs in GrL are outlined by red dotted ovals; neuron is indicated by pink arrow. (**C**)—Ventrolateral zone of GrEm: fibers of lateral cerebellar peduncles are indicated by green arrows; GC, by pink arrow. (**D**)—At the border of the ML and GrL: clusters with aNSPCs are outlined by red dotted rectangles; non-glial aNSPCs are indicated by red arrows. (**E**)—Ventrolateral part of the VC: parenchymatous clusters of aNSPCs are outlined by red dotted rectangles; superficial clusters of aNSPCs, by red dotted oval; single aNSPCs are indicated by yellow arrows. (**F**)—Ventromedial part of the VC: clusters of unmyelinated fibers are indicated by red arrows. Scale bars: (**A**,**C**)—50 µm; (**B**,**D**–**F**)—20 µm.

**Figure 4 ijms-26-11123-f004:**
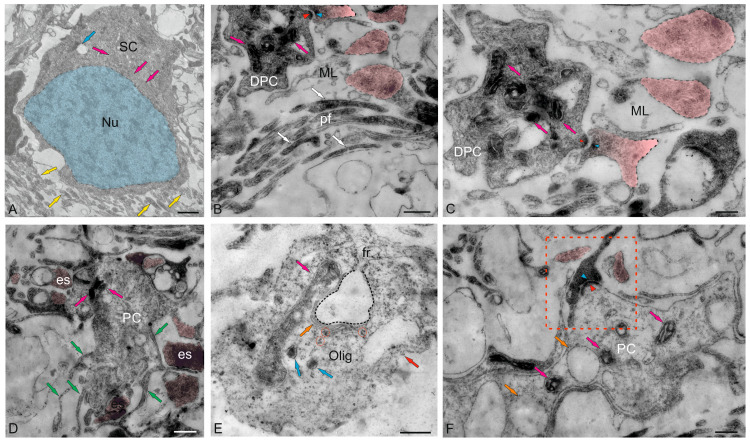
Ultrastructural organization of cells of the molecular layer (ML) of the cerebellum in juvenile chum salmon, *Oncorhynchus keta*. (**A**)—A stellate cell (SC) from the upper third of the ML: the nucleus is highlighted in blue; mitochondria are indicated by pink arrows; vacuoles, by blue arrow; secretory granules, by yellow arrows. (**B**)—A cross section at the level of the Purkinje cell dendrite (DPC) forming synaptic contacts: the presynaptic region is indicated by a red arrowhead; the postsynaptic region, by a blue arrowhead; nodular synaptic terminals, by pink arrows; mitochondria, by crimson arrows; parallel fibers (pfs), by white arrows. (**C**)—An enlarged fragment in (**B**) demonstrating the area of the primary synapse in the SCs (designations as in (**B**)). (**D**)—Fragment of the PC, receiving numerous synaptic terminals (shown in pink) of the electrotonic type (es) converge: primary dendrites of the PC are indicated by green arrows, mitochondria (pink arrows). (**E**)—Oligodendrogliocyte involved in the process of formation of myelin fibers (frs): invaginated area of the oligodendrocyte is indicated by black dotted line; large mitochondrion, by crimson arrow; oligodendrocyte process, by red arrow; rough endoplasmic reticulum (RER), by orange arrow; secretory granules with myelin are outlined by red dotted ovals; mature form of vesicles, by blue arrows. (**F**)—Structure of chemical synapse on the body of the PC (in the red dotted square): presynaptic area is indicated by red arrowhead; postsynaptic area, by blue arrowhead; synaptic endings are highlighted in pink; mitochondria, by crimson arrows; vacuoles, by orange arrows. TEM. Scale bars: (**A**)—1 µm; (**B**,**D**–**F**)—400 nm; (**C**)—200 nm.

**Figure 5 ijms-26-11123-f005:**
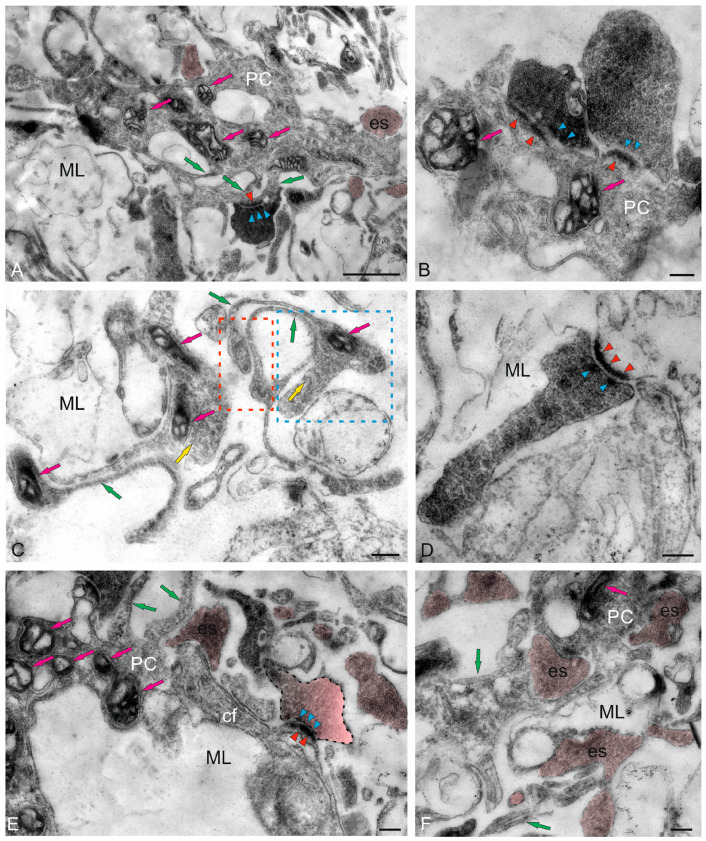
Ultrastructural organization of synaptic terminals in the molecular layer of the CC in juvenile chum salmon, *Oncorhynchus keta*. (**A**)—Primary axo-somatic synapses (430–630 nm) in the molecular layer (ML) converging on the PC, large mitochondria in the cytoplasm of the dendritic bouquet are indicated by crimson arrows; secondary dendrites of the PC are indicated by green arrows; synaptic terminals of the electrotonic type (es) are highlighted in pink; the presynaptic component in the synaptic terminal of the chemical type (1.2 μm long), by a red arrowhead; the postsynaptic component, by blue arrowheads. (**B**)—Enlarged image of a double synaptic terminal of the chemical type, converging on the primary dendrite of the PC; designations as in (**A**). (**C**)—Synaptic structures of T-shaped morphology (in the blue dotted rectangle), connected by fibrous strands (green arrows) with club-shaped endings (in the red rectangle): synaptic vesicles are indicated by yellow arrows; other designations as in (**A**). (**D**)—Enlarged fragment of a single elongated synaptic terminal (1.3 μm long); designations as in (**A**). (**E**)—On the right, patterns of synaptic terminals (shown by pink), converging on climbing fibers (cfs); on the left, a fragment of the dendritic bouquet of the PC; designations as in (**A**). (**F**)—Axo-somatic synaptic terminals of the electrotonic type, converging on the PC and adjacent areas of the ML; designations as in (**A**). TEM. Scale bars: (**A**)—1 μm; (**B**–**E**)—400 nm; (**F**)—200 nm.

**Figure 6 ijms-26-11123-f006:**
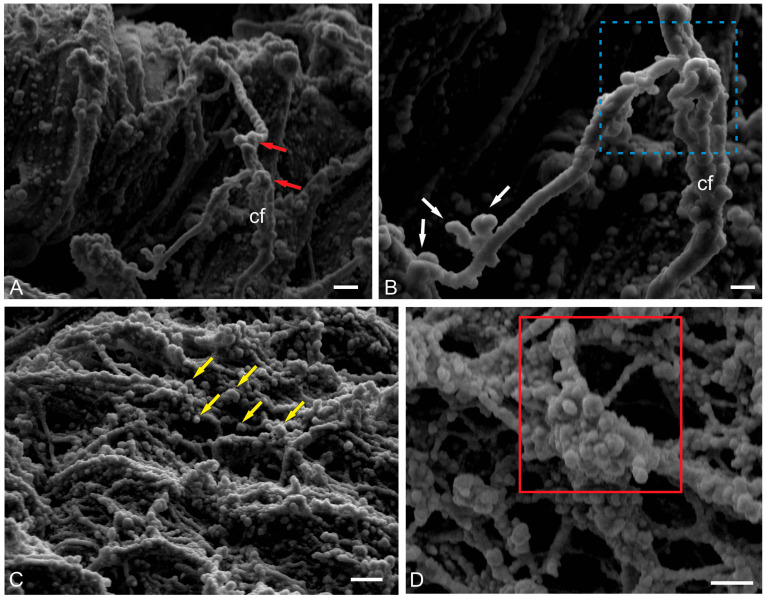
Stereoscopic organization of cf in the cerebellum of juvenile chum salmon, *Oncorhynchus keta*. (**A**)—The structure of cf afferents with varicose dilatations (red arrows) and terminal thickenings. (**B**)—A magnified fragment showing the terminal thickenings (white arrows) and the cf branching node (in the blue dotted rectangle). (**C**)—The surface over which cf extends in the cerebellum has a complex and heterogeneous relief, including numerous synaptic structures (yellow arrows). (**D**)—A magnified fragment showing synaptic areas that look like a multidimensional network with numerous spherical thickenings (in the red rectangle). SEM. Scale bars: (**A**,**C**,**D**)—2 µm; (**B**)—1 µm.

**Figure 7 ijms-26-11123-f007:**
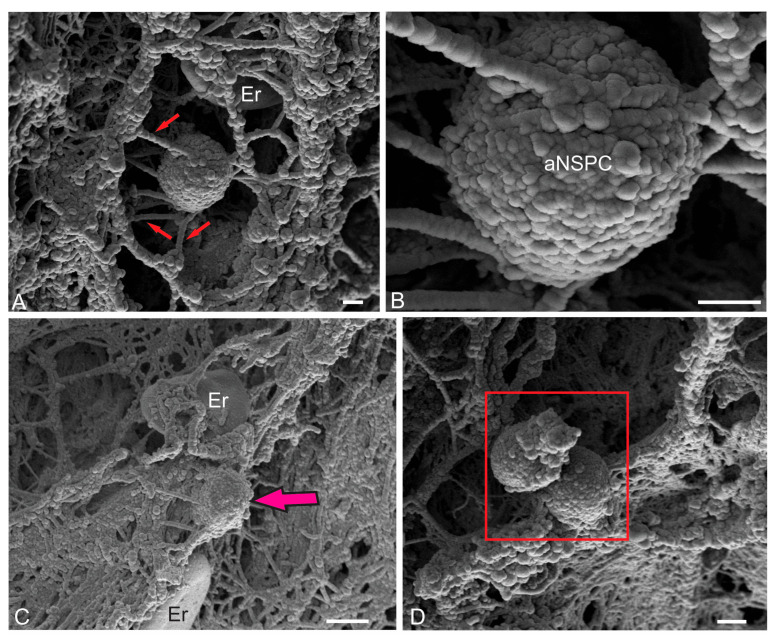
Stereoscopic organization of adult non-glial neural stem cells (aNSPCs) in the cerebellum of juvenile chum salmon, *Oncorhynchus keta*. (**A**)—aNSPCs are anchored by microfibrils (red arrows) to the surface of the ML, erythrocyte (Er). (**B**)—The surface microcytosculpture of aNSPCs is heterogeneous and has a bumpy relief. (**C**)—aNSPCs surrounded by the extracellular matrix of the pial membrane, aNSPCs (crimson arrow) were often adjacent to biconcave erythrocytes (Er) with a smooth surface. (**D**)—aNSPCs often formed paired clusters (in the red rectangle). (**E**)—Superficial stromal cluster of aNSPCs (in the red oval). (**F**)—Diffuse patterns of single aNSPC distribution (white arrow). SEM. Scale bars: (**A**,**B**,**D**)—1 µm; (**C**)—3 µm; (**E**,**F**)—2 µm.

**Figure 8 ijms-26-11123-f008:**
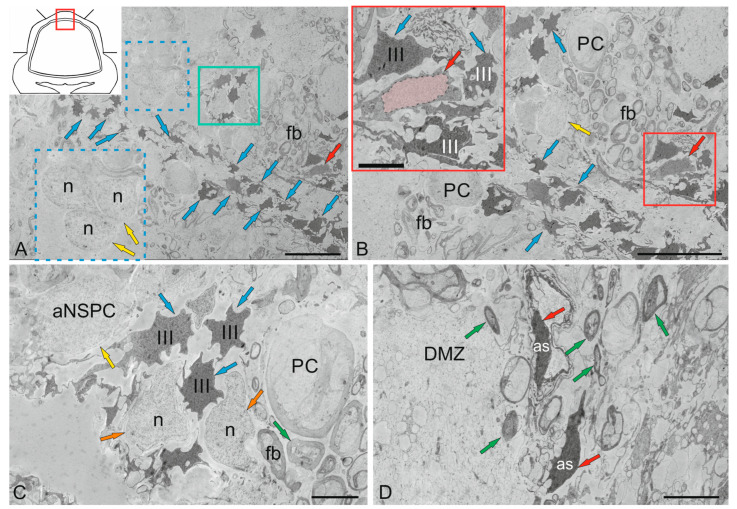
Ultrastructural organization of the dorsal matrix zone (DMZ) of the cerebellum in juvenile chum salmon, *Oncorhynchus keta*. (**A**)—General view of the dorsal matrix zone (DMZ, shown in the red square in the pictogram), including (in the dorsomedial region) an accumulation of non-glial type aNSPCs (blue dotted inset), more ventrally, a cluster of aNSPCs (in the green square): individual aNSPCs are indicated by blue arrows; patterns of sprouting myelinated fibers (fbs) and a neuroepithelial cell (NEC), by red arrow, non-glial aNSPCs (yellow arrows). (**B**)—Enlarged fragment showing details of the NEC ultrastructure (red inset) surrounded by type III aNSPCs (blue arrows) and Purkinje cells (PCs), other designations as in (**A**). (**C**)—Enlarged fragment showing ultrastructural details of the heterogeneous cluster in the green square in (**A**): type III aNSPCs are indicated by blue arrows; non-glial aNSPCs, by yellow arrow; non-glial type intermediate progenitors, by orange arrows; myelinated fibers (fbs), by green arrows; Purkinje cells (PCs). (**D**)—Organization patterns of perivascular astroglia (as) are indicated by red arrows; fibers, by green arrows. (**E**)—Non-glial aNSPCs (yellow arrows) form clusters in the ML (in the black dotted rectangle) and at the border of the GrL GrEm: type III aNSPCs are indicated by blue arrows; type IV aNSPCs, by crimson arrows; cf, by green arrows. (**F**)—Enlarged fragment in the black dotted rectangle in (**E**): non-glial aNSPCs are highlighted in yellow; aNSPCs type III (III), aNSPCs type IV (IV), secretory granules (indicated by red arrows); red dotted oval outlines the accumulation of secretory granules. TEM. Scale bars: (**A**,**B**)—20 µm; inset, (**E**)—10 µm; (**C**,**D**,**F**)—5 µm.

**Figure 9 ijms-26-11123-f009:**
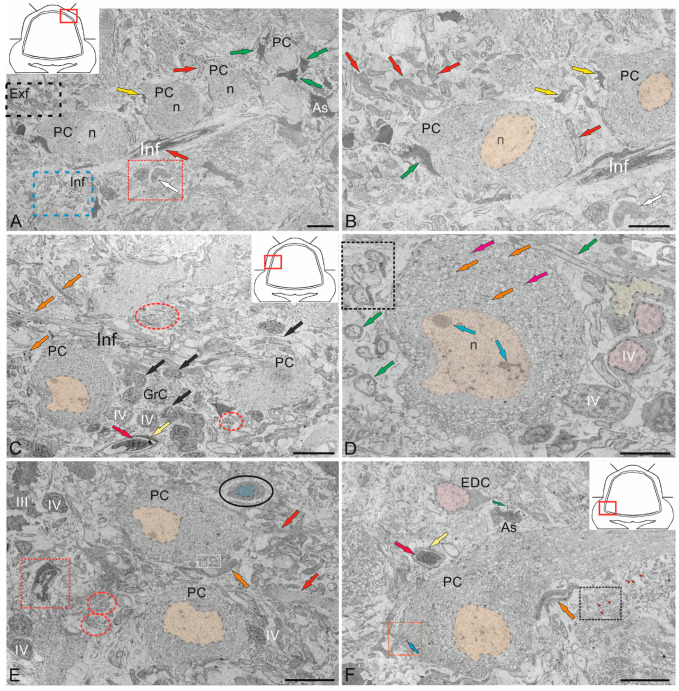
Ultrastructural organization of the ganglionic layer of the cerebellum in juvenile chum salmon, *Oncorhynchus keta*. (**A**)—General view of the ganglionic layer (GL) formed by Purkinje cells (PCs); PCs nuclei (n); fibers above the GL form the extraganglionic plexus (Exf, in the black dotted rectangle); fibers below the GL form the infraganglionic plexus (Inf, in the blue dotted rectangle); fragments of apoptotic bodies (white arrow) are outlined by the red dotted rectangle; astrocytes (As) of the protoplasmic type are indicated by green arrows; microgliocyte, by yellow arrow; infraganglionic fibers, by red arrows. (**B**)—Enlarged fragment of the GL; PCs nuclei (n) are highlighted in yellow; other designations are as in (**A**). (**C**)—GL neurons from the lateral area are ectopic in the GrL and are surrounded by type IV granular cells (black arrows); Inf fibers are indicated by orange arrows; areas with secretory granules and fibers are outlined by red dotted ovals; a vessel (yellow arrow) with an agranular macrophage is indicated by a red arrow. (**D**)—An enlarged fragment of GL in the lateral zone; nucleoli in the nucleus (n) of the PC are indicated by blue arrows; vacuoles, by orange arrows; mitochondria, by crimson arrows; fibers, by green arrows; aNSPCs type IV nuclei are highlighted in pink; a mitotically dividing nucleus IV, in yellow; a fragment containing fibers with mitochondria is outlined with a black dotted rectangle. (**E**)—Contacts of the PC with electrotonic-type fibers on the PC bodies (outlined by a white dotted rectangle), endings of cfs (orange arrows), a granulocyte (in a black oval) with large vacuoles in dense cytoplasm and a homogeneous nucleus (highlighted in blue); fragments of unbranched PC dendrites are indicated by red arrows; patterns of sprouting fibers, by red dotted ovals; patterns of apoptosis are outlined by a red dotted rectangle; type III aNSPCs—III; type IV aNSPCs—IV. (**F**)—GL neurons from the ventrolateral region: the nucleus of the ML is highlighted in yellow; the axonal hillock containing actin filaments is outlined by red dotted square (blue arrow); cf endings on the PC body are indicated by orange arrows; secretory granules are outlined by black dotted rectangle (red arrowheads); a eurydendroid cell (EDC) with an irregularly shaped nucleus (highlighted in pink); the vessel with an agranular macrophage is indicated by the yellow arrow (red arrow); an astrocyte (As) phagocytic vesicle with glutamate, by green arrow. TEM. Scale bars: (**A**–**C**,**E**,**F**)—10 µm; (**D**)—5 µm.

**Figure 10 ijms-26-11123-f010:**
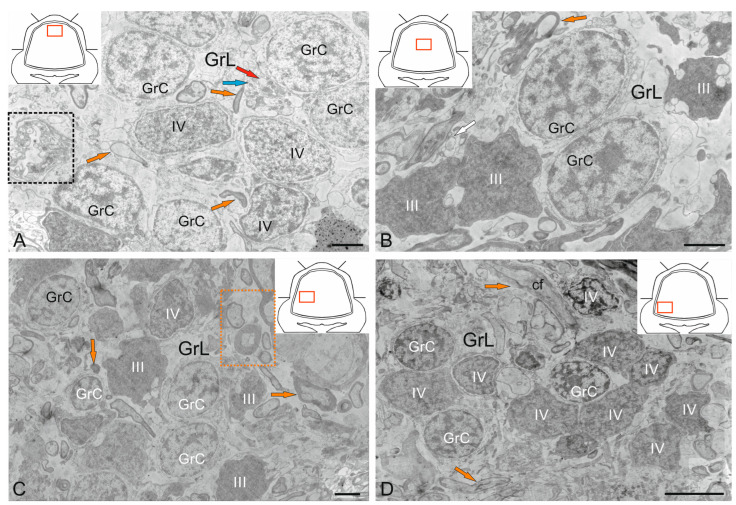
Ultrastructural organization of the granular layer of the cerebellum in juvenile chum salmon, *Oncorhynchus keta*. (**A**)—Cells of the granular layer (GrL) of the dorsomedial part of the CC: granular cells (GrC); type IV aNSPCs—IV; cfs are indicated by orange arrows; mitochondria, by blue arrows; vacuoles, by red arrows; a fragment of a lysed cell is outlined by the black dotted rectangle. (**B**)—in deeper region of GrL; type III aNSPCs—III; unmyelinated fiber is indicated by a white arrow; designations as in (**A**); (**C**)—Diffuse distribution patterns of GrL and aNSPCs types III and IV in the lateral part of the CC: myelinated fibers are indicated by orange arrows; mixed fibers are outlined by the orange dotted rectangle. (**D**)—Clustering patterns of GrL and type IV aNSPCs distribution in the ventrolateral CC and climbing fibers (cf); myelinated cf are indicated by orange arrows. (**E**)—Adult-type constitutive neurogenic niches (in the red dotted rectangle) containing type III aNSPCs (III) in the mediobasal zone of the CC; clusters of climbing fibers (cf, in the orange dotted rectangle); myelinated large cf are indicated by orange arrows, small—yellow arrows. (**F**)—Mixed adult-type neurogenic niche formed by type III and IV aNSPCs from the ventrolateral CC; designations as in (**D**,**E**). TEM. Scale bars: (**A**–**C**,**E**,**F**) 2 µm; (**D**) 5 µm.

**Figure 11 ijms-26-11123-f011:**
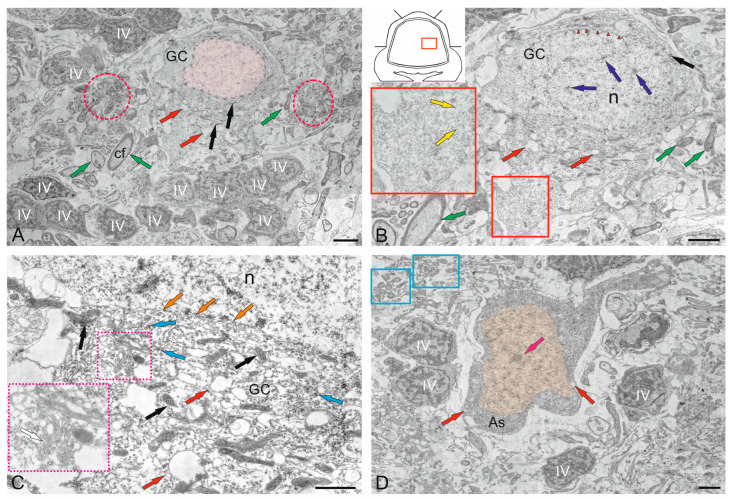
Ultrastructural organization of granular layer cells in the cerebellum of juvenile chum salmon, *Oncorhynchus keta*. (**A**)—Cells of the granular layer (GrL) of the dorsolateral part of the CC; the Golgi cell (GC) nucleus is highlighted in pink; mitochondria are indicated by black arrows; vacuoles, by red arrows; type IV aNSPCs are IV; cfs, by green arrows; nodular type cf termination patterns are outlined by red dotted ovals; (**B**)—Enlarged GC fragment: nuclear membrane boundaries are indicated by red arrowheads; heterochromatin clusters, by blue arrows; the apical cell fragment is outlined by the inset in the red rectangle; actin filaments are indicated by yellow arrows; other designations are as in (**A**). (**C**)—GC fragment at higher magnification; the boundaries of the nuclear membrane are indicated by orange arrows; ribosome clusters, by blue arrows; the inset (outlined by a magenta dotted square) shows the cisternae of the Golgi apparatus (white arrow): other designations are as in (**A**). (**D**)—Astrocyte (As); the nucleus is highlighted in orange; the nucleolus is indicated by the dark pink arrow; vacuoles, by red arrows; type IV aNSPCs—IV; the patterns of sprouting fibers are outlined by blue rectangles. (**E**)—Enlarged fragment of As contacting type IV aNSPCs (the contact area is indicated by a yellow arrow); binuclear type IV aNSPCs (nuclei are marked n); patterns of granular cell apoptosis are outlined by the black dotted rectangle; (**F**)—As fragment at higher magnification; ribosome complexes are indicated by blue arrows; actin filaments, by yellow arrows; mitochondria, by black arrows. TEM. Scale bars: (**A**)—3 µm; (**B**,**D**)—2 µm; (**C**,**E**,**F**)—1 µm.

**Figure 12 ijms-26-11123-f012:**
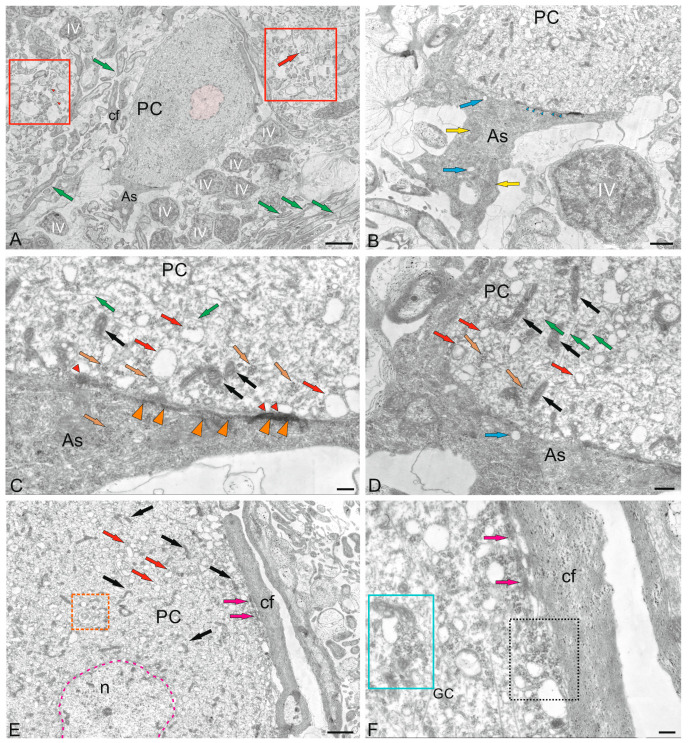
Ultrastructural organization of Purkinje cells in contact with astrocytes in the cerebellum of juvenile chum salmon, *Oncorhynchus keta*. (**A**)—Purkinje cell (PC) with nucleus (highlighted in pink) in contact with an astrocyte (As); type IV aNSPCs—IV; cf—climbing fibers (indicated by green arrows); red rectangles outline the patterns of sprouting fibers (indicated by a red arrow); extracellular glutamate granules are indicated by red arrowheads. (**B**)—Enlarged fragment of As containing rare vacuoles are indicated by blue arrows; mitochondria, by–yellow arrows; the area of contact with PC, by blue arrowheads, contact with As. (**C**)—Fragment of the contact between As and PC at higher magnification: PC vacuoles are indicated by red arrows; mitochondria, by black arrows; polyribosomes, by brown arrows; actin filaments, by green arrows; the area of contact with As, by red arrowheads (from the PC side) and orange arrowheads (from the As side). (**D**)—Fragment of the contact of As and the apical part of PC at higher magnification; vacuoles are indicated by blue arrows, other designations as in (**C**). (**E**)—Fragment of the contact of As and the basal part of PC at higher magnification: the area of contact of PC with cf is indicated by magenta arrows; the nucleus (n) is outlined by a magenta dotted line; the Golgi apparatus, by orange dotted rectangle; other designations are as in (**C**). (**F**)—Fragment of the contact of As and the basal part of PC shown in E at higher magnification; clusters of ribosomes near the plasma membrane are outlined by black dotted rectangle; clusters of ribosomes in the cytoplasm of PC, by blue rectangle; other designations are as in (**E**). TEM. Scale bars: (**A**)—3 µm; (**B**,**E**)—1 µm; (**C**,**F**)—200 nm; (**D**)—500 nm.

**Figure 13 ijms-26-11123-f013:**
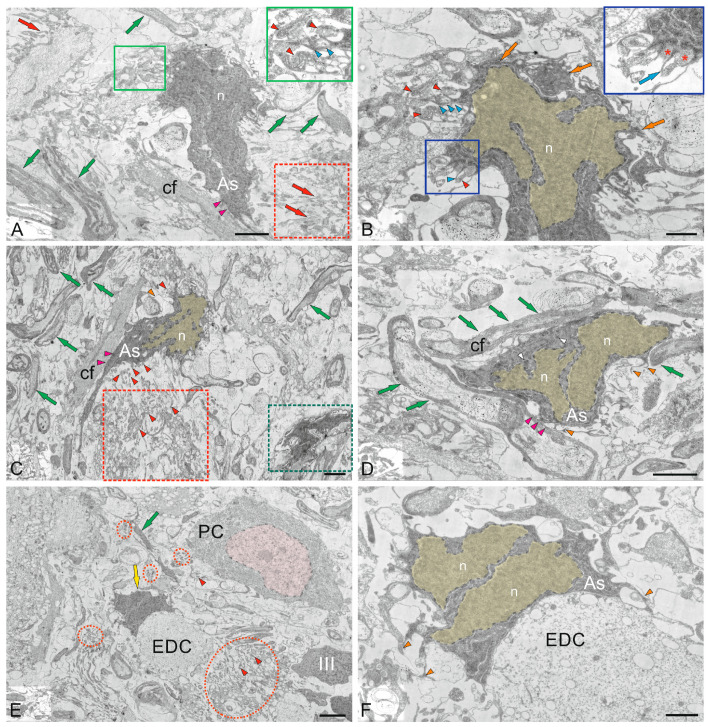
Ultrastructural organization of large astrocytes in contact with Purkinje cells in the cerebellum of juvenile chum salmon, *Oncorhynchus keta*. (**A**)—Large astrocytes located at the border of the GL and GrL: large jagged nucleus (n); inset in the green rectangle outlines astrocyte processes (blue arrowheads) surrounding membrane vesicles (red arrowheads), forming endocytic complexes; secretory granules (red arrows) are outlined by the red dotted square; fibers are indicated by green arrows; the contact zone of As and cf, by pink arrowheads. (**B**)—Enlarged fragment of As: nucleus (n) is highlighted in yellow; endocytosis is outlined by the blue inset; endocytic vesicle is indicated by red asterisk; astrocyte process is indicated by the blue arrow; As boundaries, by orange arrow; other designations are as in (**A**). (**C**)—Patterns of extracellular glutamate utilization by As: the irregularly shaped nucleus (n) is highlighted in yellow; the contact zone with cf is indicated by magenta arrowheads; the end-feet of As, by an orange arrowhead; glutamate vesicles are outlined by the red dotted rectangle; the vesicles are indicated by red arrowheads; the apoptotic cell is outlined by the green dotted rectangle. (**D**)—Fibrous-type As localized among myelinated cf (green arrows), forming tangential contacts, is indicated by magenta arrowheads; the irregularly shaped nucleus is highlighted in yellow; mitochondria are indicated by white arrowheads; the end-feet of As, by an orange arrowhead. (**E**)—Astrocytes (yellow arrow) in contact with the EDC: the PC nucleus is highlighted in pink; an accumulation of secretory granules (red arrowheads) is outlined by red dotted oval; type III aNSPCs—III. (**F**)—An enlarged fragment of As: the double nucleus is highlighted in yellow; the end-feet of As is indicated by orange arrowhead. TEM. Scale bars: (**A**,**C**,**D**)—2 µm; (**B**,**F**)—1 µm; (**E**)—3 µm.

**Figure 14 ijms-26-11123-f014:**
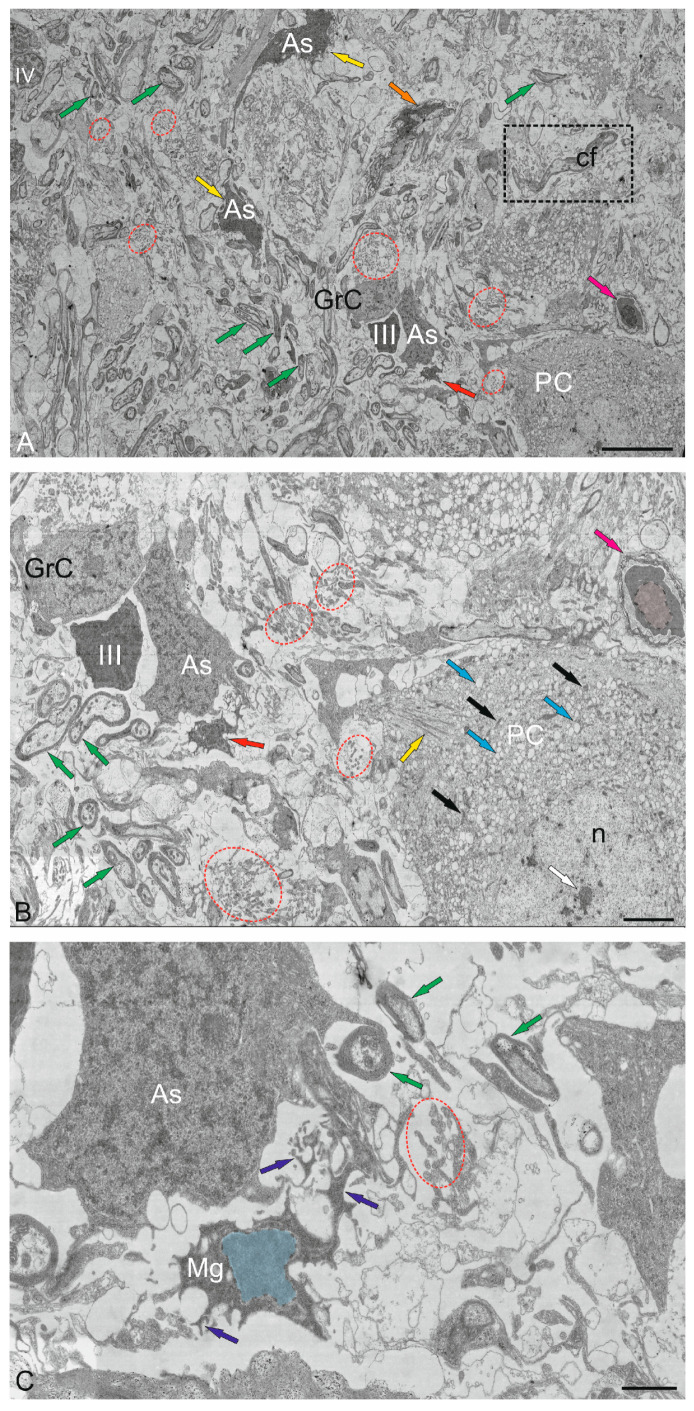
Ultrastructural organization of microglia cells in the cerebellum of juvenile chum salmon, *Oncorhynchus keta*. (**A**)—General view demonstrating localization of astrocytes (As), indicated by yellow arrows; granular cells (GrC); type III aNSPCs—III; type IV aNSPCs—IV; Purkinje cells (PC); microgliocyte is indicated by red arrow; agranular macrophage, by crimson arrow; apoptotic cell, by orange arrow; cf, by green arrow; secretory granules are outlined by red dotted ovals; cf fragment excreting secretory granules, by black dotted rectangle. (**B**)—Enlarged fragment in A: nucleolus is visible in PC nucleus (n), indicated by white arrow; actin filaments, by yellow arrow; mitochondria, by black arrow; vacuoles, by blue arrow; other designations as in (**A**). (**C**)—Fragment showing ramified microglia (Mg) and astrocyte (As) at higher magnification: the nucleus of Mg is highlighted in blue; branched processes are indicated by dark blue arrows; other designations are as in (**A**). TEM. Scale bars: (**A**)—10 µm; (**B**)—3 µm; (**C**)—1 µm.

**Table 1 ijms-26-11123-t001:** Ultrastructural characteristics of cerebellum in juvenile chum salmon, *O. keta*.

**Molecular layer**					
**Type of Cells**	**Stellate cells**			**Non-glial aNSPCs (DMZ)**	
Long axis of cell soma (µm)	8.47 ± 0.66			7.71 ± 0.45	
Short axis of cell soma (µm)	7.48 ± 0.78			6.06 ± 0.53	
Sample size	*n* = 20			*n* = 15	
**Nucleus**					
Contour	Round or oval			Round, smooth	
Long axis (µm)	6.79 ± 0.76			6.65 ± 0.36	
Short axis (µm)	5.02 ± 0.21			5.46 ± 0.54	
Chromatin	Denser euchromatin and heterochromatin			reticular euchromatin	
Color	Medium			Light	
Nucleoli	1–2 are rarely found			No or 1	
**Cytoplasm**					
Percentage/color	Medium			Light	
Mitochondria	Many			Few	
Vacuoles	Some			No	
Lipid droplets	Yes			No	
Dense bodies	No			No	
Cell contacts	No			Non-glial aNSPCs, glial aNSPCs III	
**Glial aNSPCs (GrL, ML)**					
**Type of Cells**	**III**	**IV**			**DMZ**
Long axis of cell soma (µm)	5.55 ± 0.6	5.67 ± 1.13			5.32 ± 0.92
Short axis of cell soma (µm)	3.87 ± 0.67	3.81 ± 0.42			3.07 ± 0.66
Sample size	*n* = 15	*n* = 15			*n* = 20
**Nucleus**					
Contour	elongated; irregular ± invaginations	ovoid, irregular ± invaginations			irregular ± invaginations
Long axis (µm)	4.81 ± 0.48	4.87 ± 1.21			4.73 ± 0.9
Short axis (µm)	3.39 ± 0.63	3.39 ± 0.35			2.62 ± 0.67
Chromatin	evenly distributed; non-clumped	reticulated; clumped hetero			reticulated; clumped hetero
Color	Medium	dark			Dark
Nucleoli	1–2	1 or 2, rarely visible			No visible
**Cytoplasm**					
Percentage/color	scanty, dark	scanty, medium			scanty, medium
Mitochondria	Few	few			single
Vacuoles	No	several, rarely encountered			no
Lipid droplets	0-1	no			no
Dense bodies	Yes	no			no
Localization	ML, GrL	ML, GL, GrL			ML
Cell contacts	III, IV	III, IV			III, NEC
**Granular layer**					
**Type of Cells**	**Granular cells (GrC)**	**Golgi cells (GC)**			**Astrocytes (As)**
Long axis of cell soma (µm)	6.19 ± 0.71	18.3 ± 0.81			13.9 ± 1.4
Short axis of cell soma (µm)	3.96 ± 0.53	13.02 ± 0.95			9.6 ± 0.9
Sample size	*n* = 15	*n* = 10			*n* = 10
**Nucleus**					
Contour	ovoid; elongated, smooth	ovoid; elongated, ± invaginations			ovoid; irregular ± invaginations
Long axis (µm)	5.55 ± 0.63	12.54 ± 0.98			11.3 ± 1.2
Short axis (µm)	3.52 ± 0.59	7.6 ± 0.95			8.4 ± 0.8
Chromatin	clumped heterochromatin	reticulated euchromatin, clumped heterochromatin			clumped heterochromatin
Color	Light	light			Light
Nucleoli	No	no			1
**Cytoplasm**					
Percentage/color	scanty, light, foamy	abundant/light			abundant, dark
Mitochondria	Few	many			Few
Vacuoles	Few	many			Few
Lipid droplets	No	no			No
Dense bodies	No	No			No
Cell contacts	III, IV aNSPC	Protoplasmic astrocytes			aNSPC IV
**Ganglion layer**					
**Type of Cells**	**Purkinje cells**		**Eurydendroid cells**		
Long axis of cell soma (µm)	23.28 ± 2.99		22.79 ± 0.74		
Short axis of cell soma (µm)	18.55 ± 3.22		16.91 ± 0.73		
Sample size	*n* = 10		*n* = 10		
**Nucleus**					
Contour	elongate, oval, smooth ± invaginations		oval, elongate, smooth		
Long axis (µm)	11.22 ±1.65		13.56 ± 1.83		
Short axis (µm)	8.53 ± 1.45		10.17 ± 0.69		
Chromatin	euchromatin, heterochromatin		clumped heterochromatin		
Color	Light		Light		
Nucleoli	1 or 2, rarely visible		1 or 2, rarely visible		
**Cytoplasm**					
Percentage/color	Medium		Medium		
Mitochondria	Many		Many		
Vacuoles	Few		Present		
Lipid droplets	Yes		Many		
Dense bodies	No		No		
Cell contacts	Protoplasmic astrocytes, pfs		Protoplasmic astrocytes		

## Data Availability

The original contributions presented in this study are included in the article. Further inquiries can be directed to the corresponding author.

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
