# Peer review of "Ultrastructural Characteristics of the Juvenile Chum Salmon (Oncorhynchus keta) Cerebellum: Interneuron Composition, Neuro–Glial Interactions, Homeostatic Neurogenesis, and Synaptic Plasticity"

_ijms, 2025, doi:10.3390/ijms262211123_

Round 1
Reviewer 1 Report
Comments and Suggestions for Authors
The study provides detailed ultrastructural observations of the cerebellum in juvenile chum salmon. The TEM and SEM analyses are thorough, but the manuscript would benefit from a more concise presentation that highlights the central findings.
Several points require further clarification. The discussion of “embryonization” and “neotenic status” appears speculative; these interpretations should be supported with quantitative evidence such as cell counts or synaptic density, and with comparative references.
The rationale for choosing chum salmon as the model species is not clearly explained. The authors are encouraged to elaborate on why this fish is particularly appropriate for examining cerebellar plasticity compared with other species such as salmonids, tilapia, red-tail shark, or zebrafish.
The reported absence of basket neurons is noteworthy, but was this absence established quantitatively, or could it be due to sampling limitations?
The identification of protoplasmic astrocytes is also an important point, yet it is unclear what specific criteria or markers were used to confirm their identity beyond ultrastructural appearance.
The notion of “neotenic adaptation” is intriguing, but its proposed ecological or evolutionary significance remains speculative. Comparative data from additional fish species would strengthen this argument. Similarly, the suggestion that the findings may help define targets for regenerative or anti-aging therapies seems overstated, as the evidence presented is purely morphological. This statement should be moderated.
Further clarification is also needed on several methodological and interpretive issues. How do the authors reconcile their results with reports indicating limited post-juvenile neurogenesis in zebrafish? By what ultrastructural features were apoptotic and necrotic cells distinguished, and was a method such as TUNEL used to confirm apoptosis? Since most of the specimens studied were male, could sex-related differences in neurogenesis or cerebellar structure influence the conclusions? Finally, what is the developmental equivalence of a “1 year and 2 months” juvenile salmon compared to commonly used zebrafish stages, to allow a clearer comparison across species?
Author Response
The study provides detailed ultrastructural observations of the cerebellum in juvenile chum salmon. The TEM and SEM analyses are thorough, but the manuscript would benefit from a more concise presentation that highlights the central findings.
Thank you for your valuable comment, we agree that emphasizing the main conclusions and shortening the manuscript will make it more comprehensive in scientific terms. The manuscript was shortened and the conclusions were specified in the conclusion section.
Several points require further clarification. The discussion of “embryonization” and “neotenic status” appears speculative; these interpretations should be supported with quantitative evidence such as cell counts or synaptic density, and with comparative references.
Thank you for this comment, we fully agree that quantitative data, in particular the density of distribution of cells and synaptic structures on ultrastructural sections, as well as the addition of comparative references will make it possible to clarify the terms "embryalization" and "neotenic status" to a greater extent.
In this paper, we provide quantitative data, presenting in the Morphometric Results section the most important data of verified cell types (som sizes), as well as synaptic structures. Relevant references have been added [70, 71, 72] (Bonett et al. 2022; Bruce 2013; Phung et al. 2020).
The rationale for choosing chum salmon as the model species is not clearly explained. The authors are encouraged to elaborate on why this fish is particularly appropriate for examining cerebellar plasticity compared with other species such as salmonids, tilapia, red-tail shark, or zebrafish.
Thank you for this comment, we fully agree that adding a justification for why juvenile chum salmon are particularly suitable for studying cerebellar plasticity compared to other species will make the choice of this species as a model more convincing and understandable. The relevant clarifications have been made in the Introduction section.
The reported absence of basket neurons is noteworthy, but was this absence established quantitatively, or could it be due to sampling limitations?
Thank you for this comment. The absence of basket neurons in the cerebellum of juvenile chum salmon is a sign diagnosed on a sufficiently large volume of material, and suggests the absence of basket complexes on the cell bodies in all sections of the cerebellum of juvenile chum salmon studied, which corresponds to the data on trout.
The identification of protoplasmic astrocytes is also an important point, yet it is unclear what specific criteria or markers were used to confirm their identity beyond ultrastructural appearance.
The detection of the ultrastructural identity of protoplasmic astrocytes, in particular, their conglomerates with large Purkinje cells, EDCs, as well as the identification of characteristic "terminal legs", which are a diagnostic feature of astrocytes, along with their dense cytoplasm, can be confirmed by IHC labeling of GFAP cells of similar morphology detected in juvenile chum salmon [4].
The notion of “neotenic adaptation” is intriguing, but its proposed ecological or evolutionary significance remains speculative. Comparative data from additional fish species would strengthen this argument. Similarly, the suggestion that the findings may help define targets for regenerative or anti-aging therapies seems overstated, as the evidence presented is purely morphological. This statement should be moderated.
Thank you for your comment, we fully agree that the term "neotenic adaptation", for all its potential significance, requires evidence. We certainly share the reviewer's opinion that references to other fish species could strengthen this argument. The search for practical application of the obtained data is an important task in the development of regenerative and/or anti-aging prospects for using the obtained results. Within the framework of this study, which presents morphological and ultrastructural results, the development of such areas indicates the prospects for further exploratory research.
Further clarification is also needed on several methodological and interpretive issues. How do the authors reconcile their results with reports indicating limited post-juvenile neurogenesis in zebrafish? By what ultrastructural features were apoptotic and necrotic cells distinguished, and was a method such as TUNEL used to confirm apoptosis? Since most of the specimens studied were male, could sex-related differences in neurogenesis or cerebellar structure influence the conclusions? Finally, what is the developmental equivalence of a “1 year and 2 months” juvenile salmon compared to commonly used zebrafish stages, to allow a clearer comparison across species?
We thank the Reviewer for asking these questions. Given their exceptional importance and great interest in comparative conclusions, we have added answers to these questions to the Conclusion section.
The limitation of the post-juvenile growth of zebrafish, compared with the active post-juvenile growth of juvenile chum salmon, is quite an interesting phenomenon. However, it is precisely within the framework of the proposed hypothesis of neotenic growth associated with a multiple increase in the body volume of chum salmon during the ontogenetic parallel development of the brain, and the induced activity of VSCP with peripheral localization, that this phenomenon becomes explicable. Thus, the development of this topic will allow us to provide further evidence or refutation of this hypothesis.
Apoptotic cells in the cerebellum of juvenile chum salmon have characteristic morphological and ultrastructural characteristics, in particular, compression and condensation of the cytoplasm, diagnosed in the late stages of apoptosis and their transformation into "apoptotic corpuscles". Of course, the use of molecular markers, in particular TUNEL labeling and/or the use of caspases (for example, caspase 3) as pro-apoptotic markers is an important neurobiological method to confirm such conclusions. This study does not provide data on TUNEL labeling and/or immunolabeling of caspase 3, but such data objectively exist [83]. They were obtained in our laboratory, and we have similar unpublished material. We do not have any results related to the detection of necrotic cells in the cerebellum or other parts of the brain of juvenile chum salmon. As in the case of apoptosis, the morphological phenomena of cell necrosis have clear structural and morphological features that were not detected during traumatic brain damage in juvenile chum salmon.
The question of the sex of fish (male predominance) in the studied population is very interesting, we do not exclude the possibility of the influence of sex differences in neurogenesis and the structure of the cerebellum, although this was not the purpose of this work and requires a separate special study.
The question of the equivalence of the development of juvenile salmon at the age of 1 year and 2 months, compared with the commonly used stages of the ontogenetic development of zebrafish, is certainly interesting, but in this work we did not set ourselves the goal of such a comparison. Although the life expectancy of Pacific salmon and zebrafish, despite all the numerous differences (phylogenetic position, taxonomic affiliation, etc.), is generally comparable. Both species can live up to 4-5 years, however, the body and brain volume of salmon increases thousands of times during the life cycle compared to zebrafish, which largely explains the phenomenon of "embryalization", i.e. the preservation of the "larval or neotenic status" of the salmon brain.
Reviewer 2 Report
Comments and Suggestions for Authors
The manuscript presents a comprehensive ultrastructural study of the juvenile chum salmon cerebellum using TEM and SEM. The work describes interneuron composition, glial interactions, synaptic architecture, and neural stem/progenitor cells in detail. The dataset is large, the images are valuable, and the study contributes to our understanding of teleost cerebellar organization, neurogenesis, and plasticity. The topic is of interest to readers in neurobiology, comparative anatomy, and regenerative biology.
However, the current version of the manuscript suffers from excessive length, redundancy, insufficient emphasis on novelty, and presentation issues (figures, abbreviations, language). These problems must be addressed before acceptance.
Title / Abstract
- [Title] Please highlight the novelty more clearly. The current title is descriptive but does not emphasize the innovative findings compared with previous salmonid studies.
- [Abstract, lines 8–33] The abstract is too long and contains excessive detail. Please shorten and focus on the main discoveries and their significance.
Introduction
- [Lines 36–71] Some sentences are overly long (>3 lines). Please split them into shorter, clearer sentences.
- [Line 58–60] The description “basket cells have not been found in teleosts” needs a recent reference or rephrasing to indicate uncertainty.
- [Lines 72–91] The novelty of the study compared with previous works on masou and O. mykissis not emphasized enough. Please add a clear statement of what is new in this study.
Results
- [Figures 1–3, related text] Figure legends are excessively long and duplicate text in the results. Please shorten legends to essential explanations only.
- [Throughout Section 2.1] The descriptions of cellular clusters are repetitive. Consider moving detailed morphometrics to Supplementary Materials.
- [Line 125–137] The identification of stellate cells should be supported by clearer morphological criteria or references.
Results
- [Lines 233–268] The terms “electrotonic synapses” and “chemical synapses” are used, but the definitions and criteria for distinguishing them are not clearly explained. Please clarify.
- [Figure 4–5] Some micrographs have too many colored arrows and labels, which reduce readability. Please simplify and unify the figure labeling style.
Results
- [Lines 345–394] The classification of aNSPCs into type III and IV follows Lindsay et al. (reference [33]), but the morphological/ultrastructural criteria are not clearly explained. Please provide a short description or a summary table of criteria.
- [Figures 7–8] The resolution and contrast of some SEM images are low. Please provide higher-quality images if available.
Discussion
- [Throughout Section 3] The discussion largely repeats results without sufficient interpretation. Please expand the discussion on:
Functional significance of observed synaptic diversity.
The evolutionary meaning of “neotenic status” and “embryonization.”
Relevance to neurogenesis and plasticity in other vertebrates.
- [References] Many citations are older than 15 years. Please include more recent studies (last 5 years) on teleost cerebellar neurogenesis and synaptic plasticity.
Tables and Figures
- [Tables] The manuscript repeatedly refers to “Table” or “Table 1,” but the tables are not clearly shown in the current version. Please ensure all tables are included and correctly numbered.
- [Figures] Abbreviations such as DMZ, GrEm, EDC, etc. should be listed in a glossary or abbreviation table for reader convenience.
Language and Style
- [Throughout] Numerous long and complex sentences reduce readability. Please revise for clarity and conciseness.
- [Typographical errors] Examples include “wich” → “which,” “tracke” → “track.” Please proofread carefully.
- [Line 233, 265, etc.] Replace vague terms like “variable morphology” with more precise descriptors.
Author Response
The manuscript presents a comprehensive ultrastructural study of the juvenile chum salmon cerebellum using TEM and SEM. The work describes interneuron composition, glial interactions, synaptic architecture, and neural stem/progenitor cells in detail. The dataset is large, the images are valuable, and the study contributes to our understanding of teleost cerebellar organization, neurogenesis, and plasticity. The topic is of interest to readers in neurobiology, comparative anatomy, and regenerative biology.
However, the current version of the manuscript suffers from excessive length, redundancy, insufficient emphasis on novelty, and presentation issues (figures, abbreviations, language). These problems must be addressed before acceptance.
Title / Abstract
- [Title] Please highlight the novelty more clearly. The current title is descriptive but does not emphasize the innovative findings compared with previous salmonid studies.
- [Abstract, lines 8–33] The abstract is too long and contains excessive detail. Please shorten and focus on the main discoveries and their significance.
Introduction
- [Lines 36–71] Some sentences are overly long (>3 lines). Please split them into shorter, clearer sentences.
- [Line 58–60] The description “basket cells have not been found in teleosts” needs a recent reference or rephrasing to indicate uncertainty.
- [Lines 72–91] The novelty of the study compared with previous works on masou and O. mykissis not emphasized enough. Please add a clear statement of what is new in this study.
Results
- [Figures 1–3, related text] Figure legends are excessively long and duplicate text in the results. Please shorten legends to essential explanations only.
- [Throughout Section 2.1] The descriptions of cellular clusters are repetitive. Consider moving detailed morphometrics to Supplementary Materials.
- [Line 125–137] The identification of stellate cells should be supported by clearer morphological criteria or references.
Results
- [Lines 233–268] The terms “electrotonic synapses” and “chemical synapses” are used, but the definitions and criteria for distinguishing them are not clearly explained. Please clarify.
- [Figure 4–5] Some micrographs have too many colored arrows and labels, which reduce readability. Please simplify and unify the figure labeling style.
Results
- [Lines 345–394] The classification of aNSPCs into type III and IV follows Lindsay et al. (reference [33]), but the morphological/ultrastructural criteria are not clearly explained. Please provide a short description or a summary table of criteria.
- [Figures 7–8] The resolution and contrast of some SEM images are low. Please provide higher-quality images if available.
Discussion
- [Throughout Section 3] The discussion largely repeats results without sufficient interpretation. Please expand the discussion on:
Functional significance of observed synaptic diversity.
The evolutionary meaning of “neotenic status” and “embryonization.”
Relevance to neurogenesis and plasticity in other vertebrates.
- [References] Many citations are older than 15 years. Please include more recent studies (last 5 years) on teleost cerebellar neurogenesis and synaptic plasticity.
Tables and Figures
- [Tables] The manuscript repeatedly refers to “Table” or “Table 1,” but the tables are not clearly shown in the current version. Please ensure all tables are included and correctly numbered.
- [Figures] Abbreviations such as DMZ, GrEm, EDC, etc. should be listed in a glossary or abbreviation table for reader convenience.
Language and Style
- [Throughout] Numerous long and complex sentences reduce readability. Please revise for clarity and conciseness.
- [Typographical errors] Examples include “wich” → “which,” “tracke” → “track.” Please proofread carefully.
- [Line 233, 265, etc.] Replace vague terms like “variable morphology” with more precise descriptors.
Reviewer 3 Report
Comments and Suggestions for Authors
The manuscript addresses an interesting topic that still requires extensive study given the lack of ultrastructural data on the brains of different fish species. However, there are several gaps in the work, starting from the abstract. What is the problem and the gap that the authors wish to fill? The abstract begins simply by stating that "The organization of the cerebellum of the juvenile chum salmon Oncorhynchus keta was studied using transmission (TEM) and scanning (SEM) electron microscopy. Why did the authors perform this analysis? Why did they choose the cerebellum? Are there no data in the literature? What is the usefulness and novelty of this study? Furthermore, the authors performed TEM and SEM analyses, why do they only talk about TEM results? The same conclusion reported in the abstract seems to be part of another work, the only connection with the data obtained is the presence of cells with regenerative activity, but how can they say that it is linked to environmental changes if it is not the topic of the work? I believe the abstract could be shorter and clearer if rewritten in light of the issues of climate change, adaptive differences between fish species and the gap(s) present in the literature and the great importance of the functionality and integrity of the cerebellum, an area involved in several basic functions of fish (and not only)."
Even the introduction, although providing more information, is scattered and unclear and fails to define the gaps and the issues the authors intend to address with this study: is the intent to deepen our understanding of the ultrastructure of the cerebellum of salmon or teleosts in general? Is it to identify evolutionary differences with other teleosts? Is it to define the juvenile chum salmon (O. keta) as a model for the structural and functional study of the cerebellum, particularly regarding the presence of cells with regenerative capacity? Furthermore, whatever the purpose, where is the gap in the literature? Why the choice of the juvenile chum salmon (O. keta) species?
Moving on to the results, although interesting, well illustrated and described, they require clarification regarding the analysis of the semi-thin sections: how do they manage to identify all the cell types and the structural organization at different magnifications using methylene blue?
The discussion gets lost because the authors begin with a brief, general section that seems to merely repeat what has already been reported in the literature, then divide it into subsections, the first of which discusses IHC and staining, which are not reported in the results. Specifically, the authors report "Small, densely stained, stellate-shaped cells were found in the dorsal and ventrolateral regions of the DMZ of juvenile chum salmon, O. keta (Table 1)", but Table 1 (which is also a Supplementary Table) reports measurements obtained from ultrastructural analysis, or am I mistaken? Overall, the discussion is long and confusing; I recommend making it shorter and more concise. Finally, standardize the size and style of writing.
The conclusions can be improved by specifying not only the results obtained but also their actual usefulness given the gaps in the literature. Speculating on the hypothesis that "The neotenic status observed in the juvenile chum salmon cerebellum may be part of the species' adaptation to a changing ecological niche and may help identify promising cellular and molecular targets for regenerative or anti-aging therapies in the future" is not enough to make the data obtained in this work useful and interesting, unless the problem is approached more carefully and based on literature data.
The materials and methods are fairly well described. A few more details on how the individual measurements and TEM image reconstruction were performed, and whether there are any bibliographic references the authors used to identify the different cell types and thus define the ultrastructure.
Author Response
The manuscript addresses an interesting topic that still requires extensive study given the lack of ultrastructural data on the brains of different fish species. However, there are several gaps in the work, starting from the abstract. What is the problem and the gap that the authors wish to fill? The abstract begins simply by stating that "The organization of the cerebellum of the juvenile chum salmon Oncorhynchus keta was studied using transmission (TEM) and scanning (SEM) electron microscopy. Why did the authors perform this analysis? Why did they choose the cerebellum? Are there no data in the literature? What is the usefulness and novelty of this study? Furthermore, the authors performed TEM and SEM analyses, why do they only talk about TEM results? The same conclusion reported in the abstract seems to be part of another work, the only connection with the data obtained is the presence of cells with regenerative activity, but how can they say that it is linked to environmental changes if it is not the topic of the work?
An extended ultrastructural analysis of the cerebellum of juvenile chum salmon has shown the presence of various types of interneurons in MS, GS and GrS, concerning which there is currently no complete clarity in the literature. In particular, a study of the molecular layer of the cerebellum in trout [2] did not reveal the presence of certain cell types, for example, "dark cells", and therefore, clarifying the ultrastructural organization of juvenile Pacific chum salmon can compensate for the lack of this information in the literature. Considering that currently the ultrastructural organization of the cerebellum of Pacific salmon is practically not studied, an extended ultrastructural analysis, including both transmission and scanning microscopy, seems relevant.
I believe the abstract could be shorter and clearer if rewritten in light of the issues of climate change, adaptive differences between fish species and the gap(s) present in the literature and the great importance of the functionality and integrity of the cerebellum, an area involved in several basic functions of fish (and not only)."
Thank you for the critical assessment and valuable recommendations of the esteemed reviewer. Taking into account the reviewer's comments, we conducted an audit of the abstract section and tried to take into account the useful recommendations made for improving this section.
The usefulness and novelty of this study is primarily due to the fact that, along with previous IHC studies, as well as ultrastructural analysis of adult-type precursors [4], an additional expanded ultrastructural analysis of the cerebellum of juvenile chum salmon was performed. As a result, the details of the HS organization, the cellular composition of the GrS, the synaptic structure of the MS were investigated, and an expanded ultrastructural characterization of protoplasmic type astrocytes and microglial cells was performed. The results of the IHC analysis showed the presence of various cell types labeled with PCNA, GFAP, vimentin, nestin, and HuCD [4]. The presence of various morphological cell types in juvenile chum salmon led to a more detailed ultrastructural study for an expanded characterization of cell types in the cerebellum.
In the Results and Discussion, we discuss SEM data, in particular, the results of the formation of stromal clusters on the surface of the cerebellum involved in neurogenesis and hematogenesis are discussed (pp. 324-326). The data on the structural organization of LVS identified with the help of EMS are also discussed (pp. 792-794). As part of a fairly extensive discussion, we hypothesize that the neotenic status observed in juvenile chum salmon may be part of the species' adaptation to its demanding ecological niche and will help identify promising cellular/molecular targets for regenerative or anti-aging therapy in the future.
Thank you for the recommendation about the content of the announcement section, we took into account the recommendations of the respected reviewer and made the appropriate changes.
Even the introduction, although providing more information, is scattered and unclear and fails to define the gaps and the issues the authors intend to address with this study: is the intent to deepen our understanding of the ultrastructure of the cerebellum of salmon or teleosts in general? Is it to identify evolutionary differences with other teleosts? Is it to define the juvenile chum salmon (O. keta) as a model for the structural and functional study of the cerebellum, particularly regarding the presence of cells with regenerative capacity? Furthermore, whatever the purpose, where is the gap in the literature? Why the choice of the juvenile chum salmon (O. keta) species?
Thank you for this comment. This paper actually presents data on the ultrastructural organization of the cerebellum, which provide expanded information that is currently missing and expands the understanding of the ultrastructure of the cerebellum of juvenile Pacific salmon. To date, we have not found any data that allows us to accurately identify the interneuronal composition of the cerebellum of Pacific salmon. Given the increasing volume of IHC data clarifying the phenotypic diversity of various cell types, an additional ultrastructural characteristic is relevant, which makes it possible to detail and identify the ultrastructural features of the cellular composition of the cerebellum of juvenile chum salmon.
Moving on to the results, although interesting, well illustrated and described, they require clarification regarding the analysis of the semi-thin sections: how do they manage to identify all the cell types and the structural organization at different magnifications using methylene blue?
Thank you for this comment. Morphological staining with methylene blue makes it possible to regionalize all areas of the cerebellum and identify the main anatomical and histological structures. Using this morphological coloring, it is also possible to identify different types of cells by their morphology and relationships with various anatomical layers of the cerebellum. In general, in the morphological analysis, we were guided by the general principles of the organization of the cerebellum of fish and the determination of their cellular composition, as well as the data of morphometric analysis obtained during IHC studies. We tried to present a histological "reading" of the material so that the basic structural data could be further identified using ultrastructure.
The discussion gets lost because the authors begin with a brief, general section that seems to merely repeat what has already been reported in the literature, then divide it into subsections, the first of which discusses IHC and staining, which are not reported in the results. Specifically, the authors report "Small, densely stained, stellate-shaped cells were found in the dorsal and ventrolateral regions of the DMZ of juvenile chum salmon, O. keta (Table 1)", but Table 1 (which is also a Supplementary Table) reports measurements obtained from ultrastructural analysis, or am I mistaken? Overall, the discussion is long and confusing; I recommend making it shorter and more concise. Finally, standardize the size and style of writing.
Thank you for this comment. The discussion section has been shortened and structured in accordance with the recommendations made.
The conclusions can be improved by specifying not only the results obtained but also their actual usefulness given the gaps in the literature. Speculating on the hypothesis that "The neotenic status observed in the juvenile chum salmon cerebellum may be part of the species' adaptation to a changing ecological niche and may help identify promising cellular and molecular targets for regenerative or anti-aging therapies in the future" is not enough to make the data obtained in this work useful and interesting, unless the problem is approached more carefully and based on literature data.
Thank you for this comment. The conclusion section has been improved, in particular, comments on the actual usefulness of the results obtained in this work have been added to it. We have corrected and clarified the highlighted wording, emphasizing the need for further research in this direction.
The materials and methods are fairly well described. A few more details on how the individual measurements and TEM image reconstruction were performed, and whether there are any bibliographic references the authors used to identify the different cell types and thus define the ultrastructure.
Thank you for this comment, a bibliographic reference has been added to the Materials and Methods section (Lindsey, B.W.; Darabie, A.; Tropepe, V. The cellular composition of neurogenic periventricular zones in the adult zebrafish forebrain. J. Comp. Neurol. 2012, 520, 2275-2316) on the identification of various cell types.
Round 2
Reviewer 1 Report
Comments and Suggestions for Authors
All the requested changes were done by the authors. I recommended publication of the manuscript in its present form.
Author Response
Thank you very much!
Reviewer 2 Report
Comments and Suggestions for Authors
The author addressed the comments satisfactorily.So the manuscript can be considered for the posible publication.
Author Response
Thank you very much!
Reviewer 3 Report
Comments and Suggestions for Authors
The authors did little to respond to my requests and exchanged my suggestions for inserting one or two random sentences into the abstract (not related to the introduction and discussion) without actually improving the text and its flow to make it more useful and clear. Likewise, it's unclear why this study is relevant. It certainly increases the information regarding the ultrastructure of the cerebellum of the studied species, but what purpose will this information serve? If the purpose of the work was strictly to describe the ultrastructure of the cerebellum of juvenile chum salmon, I think it would be better suited for another journal.
Author Response
Thank you for your repeated comment, but we have made significant revisions to the article and reviewed the English language.
- We have amended the abstract section based on Reviewer 3 comments; in particular, we have added information recommended by the Reviewer regarding climate change and adaptive differences in the fish brain.
- In our previous response to the Reviewer, we noted that the usefulness and novelty of this study stems primarily from the fact that, in addition to previous IHC studies and ultrastructural analysis of adult-type precursors [4], we conducted an additional, expanded ultrastructural analysis of the cerebellum of juvenile chum salmon. This allowed us to study the organizational features of the GS, the cellular composition of the GS, the synaptic structure of the MS, and conduct an expanded ultrastructural characterization of protoplasmic astrocytes and microglial cells. The IHC results revealed the presence of various cell types labeled with PCNA, GFAP, vimentin, nestin, and HuCD [4]. The presence of various morphological cell types in juvenile chum salmon necessitated a more detailed ultrastructural study to further characterize cerebellar cell types. Apparently, the reviewer did not read the response to his comments regarding this section.
- In the Results and Discussion section, we discuss the SEM data, in particular, the formation of stromal clusters on the cerebellar surface involved in neurogenesis and hematopoiesis (pp. 324–326). We also discuss the structural organization of the cerebellar system, as revealed by EMS (pp. 792–794). In this extensive discussion, we hypothesize that the neotenic status observed in juvenile chum salmon may be part of the species' adaptation to its complex ecological niche and will help identify promising cellular and molecular targets for regenerative or anti-aging therapies in the future. The purpose of this paper is clearly stated: The aim of this work was to study the ultrastructural organization of the cerebellum of juvenile chum salmon O. keta in the context of interneuron composition, neuro-glial relationships, homeostatic neurogenesis, and synaptic plasticity.
- We do not consider it appropriate to change the structure of the text, as other reviewers agreed with the changes, finding them understandable and useful. The relevance of this work is described in the introduction. No specific recommendations for improving the text were provided in the second review. We have made corrections in accordance with the recommendations of all reviewers. Following the publication of the preprint, approximately 30 journals expressed interest in publishing this work, so if the Editor decides to reject this article, we are prepared to resubmit it to other journals.

Round 3
Reviewer 3 Report
Comments and Suggestions for Authors
The authors explained their responses and their purpose better. Although they used an inappropriate, disrespectful, and offensive tone, since they're dealing with a professional, they won't have any problems this time. I advise them to avoid responses like "Following the publication of the preprint, approximately 30 journals expressed interest in publishing this work, so if the Editor decides to reject this article, we are prepared to resubmit it to other journals." This will avoid denigrating the journal that is giving you a chance. Above all, remember that even if many journals (often lower-ranking than the one you selected) ask for a preprint, it doesn't mean it's a good, well-written work; it simply means they're looking for work to publish, and they've only read the abstract of the preprint, and no competent reviewer has checked the entire work.